# TCF7L1 promotes skin tumorigenesis independently of β-catenin through induction of LCN2

Amy T Ku[1,2,3], Timothy M Shaver[1,2], Ajay S Rao[1,2], Jeffrey M Howard[1,2], Christine N Rodriguez[1,2,4], Qi Miao[1,2], Gloria Garcia[1,2], Diep Le[1,2], Diane Yang[1,2,4], Malgorzata Borowiak[1,2,4,5,6], Daniel N Cohen[7], Vida Chitsazzadeh[8], Abdul H Diwan[9], Kenneth Y Tsai[10,11], Hoang Nguyen[1,2,3,4,5,9,12]*

[1]Stem Cell and Regenerative Medicine Center, Baylor College of Medicine, Houston, United States; [2]Center for Cell and Gene Therapy, Baylor College of Medicine, Houston, United States; [3]Interdepartmental Program in Translational Biology and Molecular Medicine, Baylor College of Medicine, Houston, United States; [4]Department of Molecular and Cellular Biology, Baylor College of Medicine, Houston, United States; [5]Program in Developmental Biology, Baylor College of Medicine, Houston, United States; [6]McNair Medical Institute, Baylor College of Medicine, Houston, United States; [7]Department of Pathology and Immunology, Michael E. DeBakey VA Medical Center, Baylor College of Medicine, Houston, United States; [8]Department of Translational Molecular Pathology, University of Texas MD Anderson Cancer Center, Houston, United States; [9]Department of Dermatology, Baylor College of Medicine, Houston, United States; [10]Department of Tumor Biology, Moffitt Cancer Center, Tampa, United States; [11]Department of Anatomic Pathology, Moffitt Cancer Center, Tampa, United States; [12]Dan L. Duncan Cancer Center, Baylor College of Medicine, Houston, United States

*For correspondence: hoangn@bcm.edu

Competing interests: The authors declare that no competing interests exist.

**Abstract** The transcription factor *TCF7L1* is an embryonic stem cell signature gene that is upregulated in multiple aggressive cancer types, but its role in skin tumorigenesis has not yet been defined. Here we document TCF7L1 upregulation in skin squamous cell carcinoma (SCC) and demonstrate that TCF7L1 overexpression increases tumor incidence, tumor multiplicity, and malignant progression in the chemically induced mouse model of skin SCC. Additionally, we show that downregulation of TCF7L1 and its paralogue TCF7L2 reduces tumor growth in a xenograft model of human skin SCC. Using separation-of-function mutants, we show that TCF7L1 promotes tumor growth, enhances cell migration, and overrides oncogenic RAS-induced senescence independently of its interaction with *β*-catenin. Through transcriptome profiling and combined gain- and loss-of-function studies, we identified LCN2 as a major downstream effector of TCF7L1 that drives tumor growth. Our findings establish a tumor-promoting role for TCF7L1 in skin and elucidate the mechanisms underlying its tumorigenic capacity.

## Introduction

Cancer cells have many characteristics in common with stem cells (*Beck and Blanpain, 2013*). When the expression profiles of several types of poorly differentiated and aggressive tumors were compared to embryonic stem (ES) cell signature genes, a subset of ES cell transcriptional regulators was

found to be highly enriched (*Ben-Porath et al., 2008*). One such ES cell signature gene enriched in aggressive tumors is *TCF7L1* (also known as *TCF3*).

TCF7L1 is a member of the LEF/TCF (lymphoid enhancer factor/T cell factor) family of transcription factors, which act as DNA binding partners for the WNT mediator β-catenin (*Korinek et al., 1998*). LEF/TCF proteins contain a highly conserved DNA binding high-mobility group (HMG)-box, a conserved β-catenin binding domain in their amino terminus, and a Groucho/TLE binding domain in their central region (*Arce et al., 2006*). In response to canonical WNT signaling, β-catenin is translocated to the nucleus and partners with LEF/TCF members to activate transcription of WNT target genes (*Behrens et al., 1996*; *Molenaar et al., 1996*; *van de Wetering et al., 1997*). In the absence of WNT ligands, LEF/TCF binds to Groucho/TLE co-repressors to inhibit transcription of WNT target genes (*Cavallo et al., 1998*; *Roose et al., 1998*).

TCF7L1 is expressed in ES cells and adult progenitors of many tissues, including hematopoietic (*Ivanova et al., 2002*; *Chambers et al., 2007*), neural (*Ivanova et al., 2002*; *Lie et al., 2005*) and skin (*DasGupta and Fuchs, 1999*; *Merrill et al., 2001*). In skin, TCF7L1 is expressed in embryonic epidermal cells, and postnatally its expression is restricted to hair follicle stem cells and their immediate progeny, the hair germ and the outer root sheath cells (*DasGupta and Fuchs, 1999*; *Merrill et al., 2001*; *Nguyen et al., 2006*). Overexpression of TCF7L1 in the newborn epidermis blocks terminal differentiation and maintains cells in an undifferentiated state (*Merrill et al., 2001*; *Nguyen et al., 2006*). In the epidermis, the expression pattern of TCF7L1 is mirrored by its closely related paralogue TCF7L2 (also known as TCF4), which appears to possess similar functions and can compensate for the absence of TCF7L1. Loss of both TCF7L1 and TCF7L2 in skin results in hair follicle morphogenesis defects, as well as failure of long-term proliferation/self-renewal and wound repair (*Nguyen et al., 2009*). Recently, we found that TCF7L1 is induced at the skin wound edge and its overexpression promotes epidermal cell migration and wound healing (*Miao et al., 2014*).

Although TCF7L1 can bind to β-catenin, a number of its critical developmental functions occur through its role as a transcriptional co-repressor with Groucho/TLE (*Wray et al., 2011*; *Yi et al., 2011*; *Kim et al., 2000*; *Pereira et al., 2006*; *Gribble et al., 2009*; *Wu et al., 2012*; *Miao et al., 2014*). For instance, although *TCF7L1*-null mice die embryonically with gastrulation defects (*Merrill et al., 2004*), mice homozygously expressing a knock-in *TCF7L1ΔN* mutant that does not bind to β-catenin gastrulate normally (*Wu et al. 2012*), suggesting that TCF7L1's role in β-catenin binding and canonical WNT activation is not essential in this context. However, the knock-in *TCF7L1ΔN* mutant mice die at birth with multiple developmental defects, suggesting that TCF7L1 requires binding to β-catenin to allow normal development to occur in other tissues. In ES cells, WNT signaling activation does lead to the interaction of β-catenin with TCF7L1; however, rather than forming a transcriptional activation complex, β-catenin instead stimulates TCF7L1's removal from the promoters of pluripotency genes to allow their derepression (*Wray et al., 2011*; *Yi et al., 2011*). In addition, there is evidence that WNT signaling actually downregulates TCF7L1 expression in ES cells (*Atlasi et al., 2013*; *Shy et al., 2013*) and that binding to β-catenin stimulates TCF7L1 degradation (*Shy et al., 2013*). TCF7L1 downregulation by WNT is also observed in neural progenitor cells (*Kuwahara et al., 2014*). Together, these data suggest that WNT signaling is unlikely to stimulate transcription of WNT target genes through the formation of an activating β-catenin/TCF7L1 complex. However, a study in breast cancer cells showed that *TCF7L1* knockdown led to the simultaneous upregulation and downregulation of different subsets of WNT target genes, suggesting that TCF7L1 may directly or indirectly play an activating role in WNT signaling (*Slyper et al., 2012*).

In human breast cancer, high levels of TCF7L1 are found in high-grade tumors and its expression is associated with poor survival (*Slyper et al., 2012*). Importantly, downregulation of *TCF7L1* was shown to decrease tumor growth and reduce metastasis rate (*Slyper et al., 2012*). However, the mechanism underlying TCF7L1's tumor-promoting role in breast cancer remains to be defined. In colorectal cancer, high level of *TCF7L1* mRNA also correlates with shorter survival of patients (*Murphy et al., 2016*). Knocking out TCF7L1 reduced growth of a colorectal tumor cell line in vitro and reduced the size of xenografted tumors (*Murphy et al., 2016*). EPHB3 was among the genes upregulated in TCF7L1-null cells, but its knockdown only partially rescued the growth defect of TCF7L1-null cells in vitro, suggesting that other downstream effectors of TCF7L1 are needed to execute the full function of TCF7L1.

In skin squamous cell carcinoma (SCC), β-catenin is essential for tumor growth both in a chemically-induced mouse model of skin SCC and a xenograft model of human skin SCC (*Malanchi et al., 2008*). Coincidentally, *Tcf7l1* mRNA is highly expressed in mouse papillomas (*Malanchi et al., 2008*), which are premalignant lesions that precede SCC. However, it is unclear what function TCF7L1 plays in the development of SCC, and whether its tumor-promoting role requires its interaction with β-catenin. Intriguingly, TRANSFAC-based motif analysis identified TCF7L1 as one of the 17 transcription factors whose targets are significantly altered between normal skin and premalignant tumors (papilloma/actinic keratosis) as well as between premalignant and malignant skin tumors (SCC) in both human and mice (*Chitsazzadeh et al., 2016*).

In this study, we show that overexpression of TCF7L1 increases tumor incidence, multiplicity, and malignant conversion in a mouse model of skin SCC. In a xenograft model of human skin SCC, we show that overexpression of TCF7L1 also promotes tumor growth while downregulation of TCF7L1 and its paralogue TCF7L2 decreases tumor growth. Moreover, we demonstrate that TCF7L1 overexpression promotes tumor growth, enhances migration, and suppresses oncogenic RAS-induced senescence independently of β-catenin interaction. Finally, we identified the secreted protein LCN2 as the downstream effector of TCF7L1 that stimulates tumor growth. LCN2 (lipocalin 2, also known as NGAL, 24p3, uterocalin, or siderocalin) is a secreted protein that is induced in response to stress and injury and is overexpressed in many types of cancer (*Li and Chan, 2011*; *Rodvold et al., 2012*). Our findings establish a causal, β-catenin independent role for TCF7L1 in skin tumorigenesis and shed light on the mechanisms underlying its tumorigenic function.

## Results

### TCF7L1 is upregulated in papillomas and skin SCC

Development of human SCC typically begins with a premalignant lesion, actinic keratosis, which then progresses into invasive SCC (*Alam and Ratner, 2001*). Similarly, in the well-established murine two-stage DMBA/TPA induced skin SCC model, where mice are given a single dose of mutagenic 9,10-dimethyl-1,2-benzanthracene (DMBA) followed by a twice weekly dose of the proinflammatory agent 12-O-tetradecanoyl phorbol-13-acetate (TPA), mice first develop premalignant squamous papillomas, and a portion of these progress to SCC (*Abel et al., 2009*). A recent whole-exome sequencing study showed that DMBA/TPA-induced skin SCCs share a number of commonly mutated genes with human SCC, confirming the relevance of this mouse model (*Nassar et al., 2015*).

To assess the role of TCF7L1 in skin SCC, we first evaluated the expression pattern of TCF7L1 in DMBA/TPA-induced mouse papillomas and skin SCC. Although TCF7L1 expression is normally restricted to the bulge and outer root sheath cells of the hair follicle and is absent in the interfollicular epidermis, we observed widespread expression in papillomas and even greater expression in skin SCC (*Figure 1A*). In the UV-induced mouse model of skin SCC, where papillomas and SCC develop from skin that is chronically exposed to UV radiation, increased expression of TCF7L1 was also observed in 2 out of 4 cases of skin SCC examined (*Figure 1B*).

### Overexpression of TCF7L1 promotes tumor development and progression in a chemically induced mouse model of skin SCC

In mice subjected to the two-stage DMBA/TPA chemical carcinogenesis protocol (*Yuspa et al., 1996*), a majority of cases show DMBA-induced activating mutations in the endogenous *H-Ras* proto-oncogene (*Quintanilla et al., 1986*). Subsequently, TPA promotes inflammation which induces tumor growth by promoting clonal expansion of the mutated cells (*Balmain et al., 1988*). One major advantage of this chemical carcinogenesis model is that the initiation and promotion steps are distinct; using a well-established dosing scheme, neither agent alone is sufficient for tumor formation, allowing us to determine which step (tumor initiation or promotion) can be substituted for by TCF7L1 overexpression.

We employed the DMBA/TPA carcinogenesis protocol on our previously generated tet-inducible *Tcf7l1* mice. The tet-inducible *Tcf7l1* mice were engineered to express two transgenes, *KRT14-rtTA* and *TRE-Tcf7l1* (*Nguyen et al., 2006*). The *KRT14-rtTA* transgene expresses the tetracycline-sensitive transactivator, *rtTA-VP16* (*Urlinger et al., 2000*; *Knott et al., 2002*) under the control of the *Keratin 14 (KRT14)* promoter, which is active in epidermal basal cells and the outer root sheath of

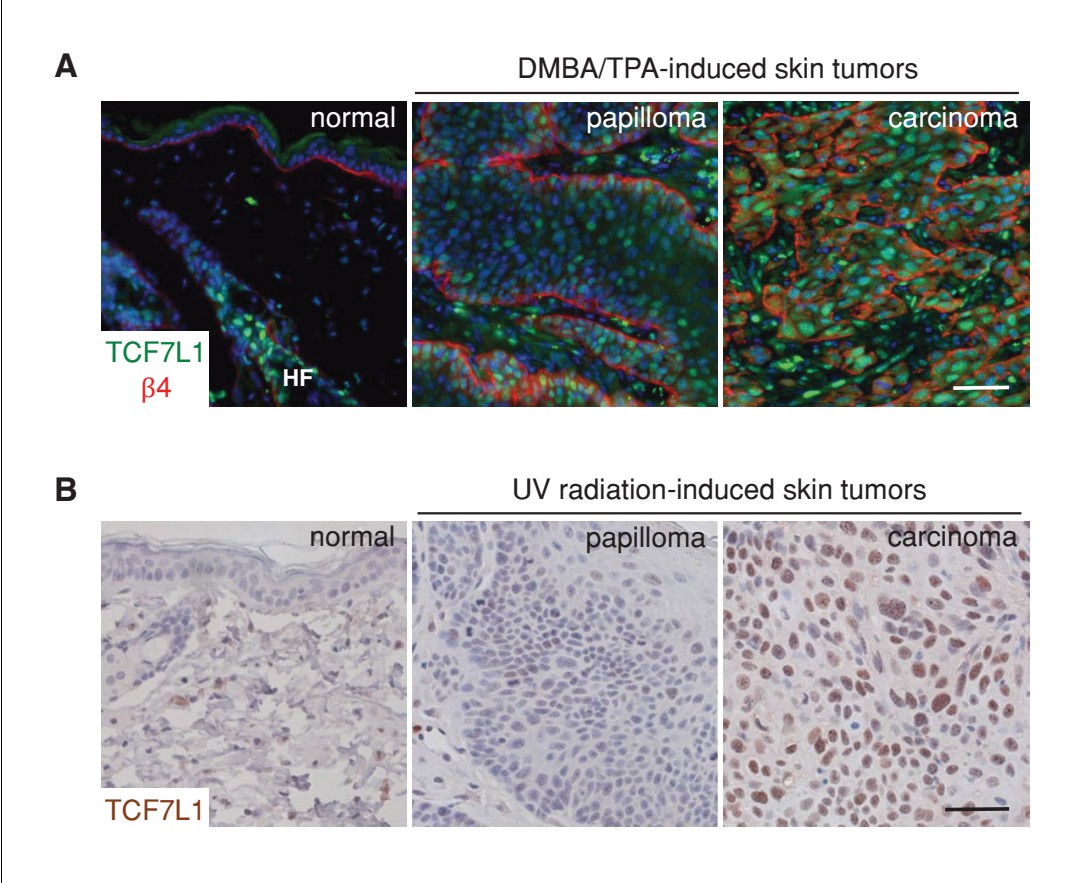

**Figure 1.** TCF7L1 is overexpressed in papillomas and skin SCC. (**A**) Immunofluorescence analyses of frozen sections of normal skin, papilloma and SCC induced by DMBA/TPA using antibodies against TCF7L1 (green) and β4-integrin (red). (**B**) Immunohistochemistry of untreated abdomen epidermis and UV-induced papilloma and SCC with antibody against TCF7L1. Bar denotes 50 μm.

the hair follicle (*Nguyen et al., 2006*). The *TRE-Tcf7l1* transgene expresses myc-epitope tagged *Tcf7l1* under the control of a tetracycline regulatory element (TRE). Mice containing these two transgenes exhibit strict tetracycline (doxycycline, Dox)-dependent expression of myc-TCF7L1 (*Nguyen et al., 2006*).

Previously, we showed that induction of TCF7L1 in newborn mice blocks differentiation in the stratified epidermis, sebaceous gland, and hair follicle. Here we found that prolonged overexpression of TCF7L1 in adult skin results in hyperproliferation of the stratified epidermis and hair follicle (*Figure 2—figure supplement 1*). Whereas expression of the proliferative marker Ki67 did not change after five days of TCF7L1 induction, Ki-67 was detected throughout the interfollicular epidermis and the hair follicle after one month of TCF7L1 induction. In the suprabasal layer of the TCF7L1-induced epidermis, we found aberrant expression of Keratin 6 (KRT6), which normally marks the inner layer of the hair follicle but is also induced in hyperproliferative cells at the wound edge, as well as hyperplastic, neoplastic, and psoriatic epidermis (*Moll et al., 1982*; *Weiss et al., 1984*; *Stoler et al., 1988*). After one month of TCF7L1 induction, hair follicle cells in the TCF7L1-induced skin are more proliferative as shown by increased Ki67 staining and altered hair follicle morphology (*Figure 2—figure supplement 1*). In contrast to the quiescent and short hair follicles in the control skin, the hair follicles in TCF7L1-induced skin are elongated, illustrating growth.

In order to determine the role of TCF7L1 in skin tumorigenesis, we put the tet-inducible *Tcf7l1* mice (*KRT14-rtTA;TRE-Tcf7l1*) and their littermate controls (*KRT14-rtTA* or *TRE-Tcf7l1*) on a doxycycline-containing diet one week prior to DMBA/TPA treatment (*Figure 2A*). Under a high-dose DMBA (100 nmol) and TPA (10 nmol) treatment, TCF7L1-induced mice developed a greater number

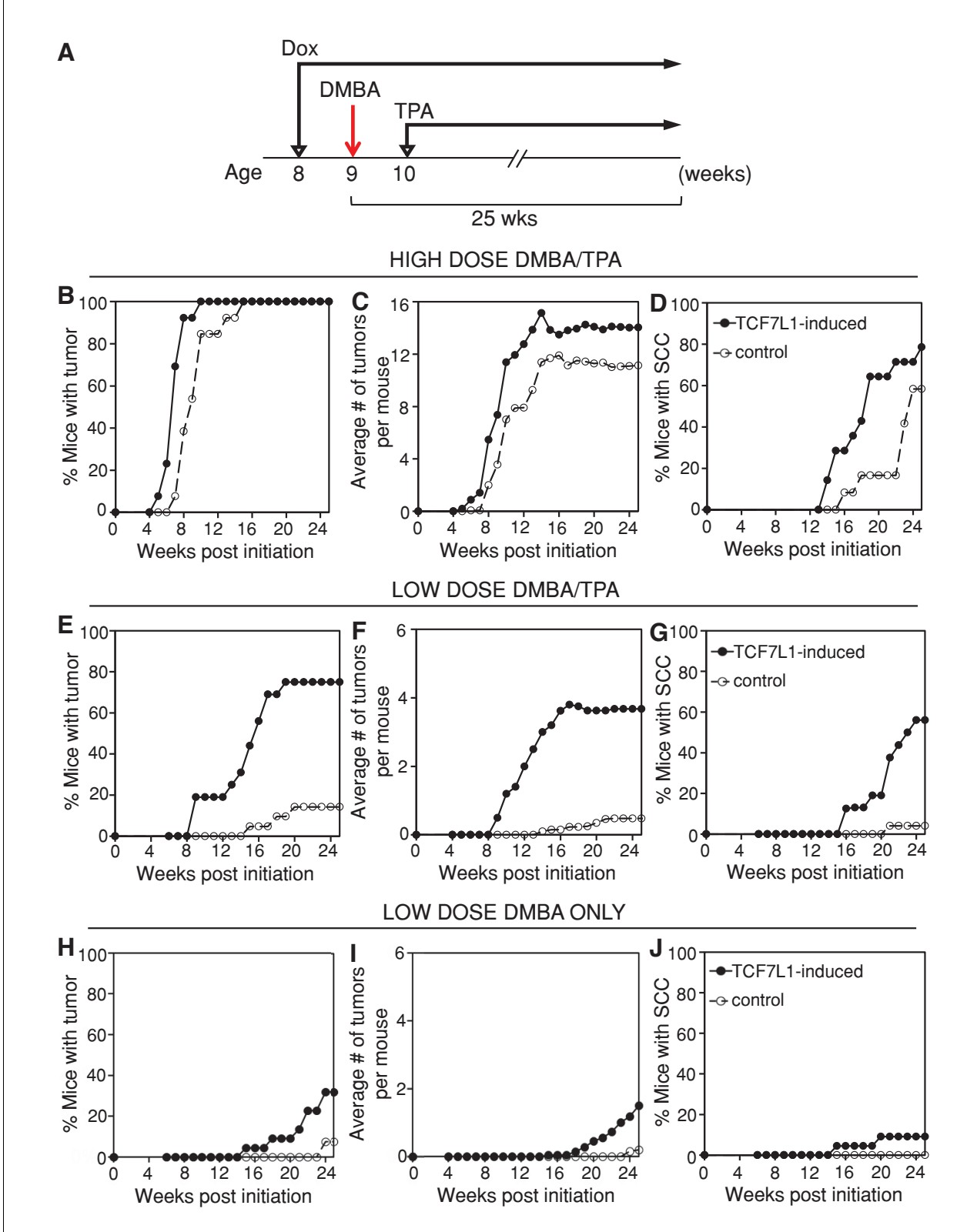

**Figure 2.** TCF7L1 overexpression increases tumor incidence and multiplicity and accelerates malignancy progression in the mouse model of skin SCC. (A) Experimental scheme testing the role of TCF7L1 in tumorigenesis. Eight-week-old tet-inducible *Tcf7l1* (*KRT14-rtTA;TRE-Tcf7l1*) and control (*KRT14-rtTA* or *TRE-Tcf7l1*) mice were put on a doxycycline-containing diet one week prior to DMBA treatment and for the remainder of the experiment. Shaved backskins were topically treated once with DMBA and then twice weekly with TPA for 25 weeks. (B–D) Effect of TCF7L1 overexpression under a

*Figure 2 continued on next page*

*Figure 2 continued*

high dose of DMBA (100 nmol) and TPA (10 nmol) on skin tumorigenesis. n = 17 (*KRT14-rtTA;TRE-Tcf7l1*), n = 16 (*KRT14-rtTA* or *TRE-Tcf7l1*). (**B**) Tumor incidence. **p<0.01. (**C**) Average number of tumors per mouse. (**D**) Percentage of mice with SCC. p=0.08. (**E–G**) Effect of TCF7L1 overexpression under a low dose of DMBA (25 nmol) and TPA (1 nmol) on skin tumorigenesis. n = 16 (*KRT14-rtTA;TRE-Tcf7l1*), n = 21 (*KRT14-rtTA* or *TRE-Tcf7l1*). (**E**) Tumor incidence. ***p<0.001. (**F**) Average number of tumors per mouse. (**G**) Percentage of mice with SCC. ***p<0.001. (**H–J**) Effect of TCF7L1 overexpression on skin tumorigenesis following a low dose of DMBA without TPA treatment. 8-week-old tet-inducible *Tcf7l1* (*KRT14-rtTA;TRE-Tcf7l1*) and control (*KRT14-rtTA* or *TRE-Tcf7l1*) mice were put on a doxycycline-containing diet one week prior to DMBA treatment and for the remainder of the experiment. Shaved backskins were topically treated once with low dose DMBA (25 nmol) and then twice weekly with an acetone vehicle control in place of TPA. n = 22 (*KRT14-rtTA; TRE-Tcf7l1*), n = 26 (*KRT14-rtTA* or *TRE-Tcf7l1*). (**H**) Tumor incidence. TCF7L1-induced versus control: *p<0.05. (**I**) Average number of tumors per mouse. (**J**) Percentage of mice with SCC. TCF7L1-induced versus control: p=0.1198.

The following figure supplements are available for figure 2:

**Figure supplement 1.** Prolonged induction of TCF7L1 in adult skin results in hyperproliferation.

**Figure supplement 2.** Myc epitope-tagged TCF7L1 is induced in skins and tumors of tet-inducible mice on a doxycycline-containing diet.

**Figure supplement 3.** Tet-inducible TCF7L1 is tightly regulated and not sufficient to promote tumor formation.

of papillomas at a faster rate than control mice did (*Figure 2B,C*; n = 17 TCF7L1-induced, n = 16 control). Progression to SCC also occurred at a higher frequency and faster rate in the TCF7L1-induced animals (*Figure 2D*). While 100% of animals developed papillomas under the high-dose regimen, the effect of TCF7L1 overexpression was much more evident upon a low dose of DMBA (25 nmol) and TPA (1 nmol). In contrast to control animals that experienced a lower than 20% frequency of tumor formation, 80% of TCF7L1-induced animals formed papilloma and 60% developed SCC (*Figure 2E–G*; n = 16 TCF7L1-induced, n = 21 control). The size of the tumors, however, did not differ significantly between the control and TCF7L1-induced groups (data not shown).

We confirmed that the myc-epitope tagged TCF7L1 transgenic protein was induced in the adult skin and tumors of TCF7L1-induced mice (*Figure 2—figure supplement 2*). In addition, the increased tumor formation observed in TCF7L1-induced mice was tightly tet-regulatable; when not fed doxycycline-containing chow, TCF7L1-inducible mice subjected to DMBA/TPA treatment were indistinguishable from their control littermates, developing tumors at a comparable rate (*Figure 2—figure supplement 3*; n = 17 TCF7L1-uninduced, n = 16 control).

To determine whether TCF7L1 overexpression alone was sufficient to promote tumor formation, we treated TCF7L1-induced mice with an acetone vehicle control instead of DMBA/TPA. Under this condition, TCF7L1-induced mice did not spontaneously develop any tumors during the 25 week regimen (*Figure 2—figure supplement 3*; n = 20 TCF7L1-induced, n = 29 control). To determine whether TCF7L1 could substitute for TPA's tumor-promoting role, we treated the TCF7L1-induced mice with an initial dose of DMBA (25 nmol) and an acetone vehicle control in place of TPA. We found that while almost none of the control mice developed tumors, over 30% of the TCF7L1-induced mice developed papilloma (*Figure 2H–J*; n = 22 TCF7L1-induced, n = 26 control). That the DMBA/acetone treated TCF7L1-induced mice developed tumors at a similar rate to the DMBA/TPA treated control mice (*Figure 2E–J*) suggests that TCF7L1 can act as a tumor promoter, since its expression was able to substitute for pro-inflammatory TPA application in the DMBA/TPA chemical carcinogenesis protocol.

## Downregulation of TCF7L1 and TCF7L2 reduces tumor growth in a xenograft model of human skin SCC

We found that *TCF7L1* is overexpressed in a majority of human skin SCC cell lines (*Figure 3A*). To evaluate the role of TCF7L1 in human skin SCC, we sought to assess the effect of downregulating *TCF7L1* and also *TCF7L2* in a xenograft model, since *Tcf7l2* can compensate for *Tcf7l1* loss of function in murine skin development (*Nguyen et al., 2006*). We first identified two different sets of shRNAs against *TCF7L1* and *TCF7L2* that efficiently downregulate their mRNA and protein expression levels (*Figure 3B,C*) in a well-characterized human skin SCC cell line, SCC13 (*Rheinwald and Beckett, 1981*). We next transduced and selected SCC13 cells that express tet-inducible shRNA

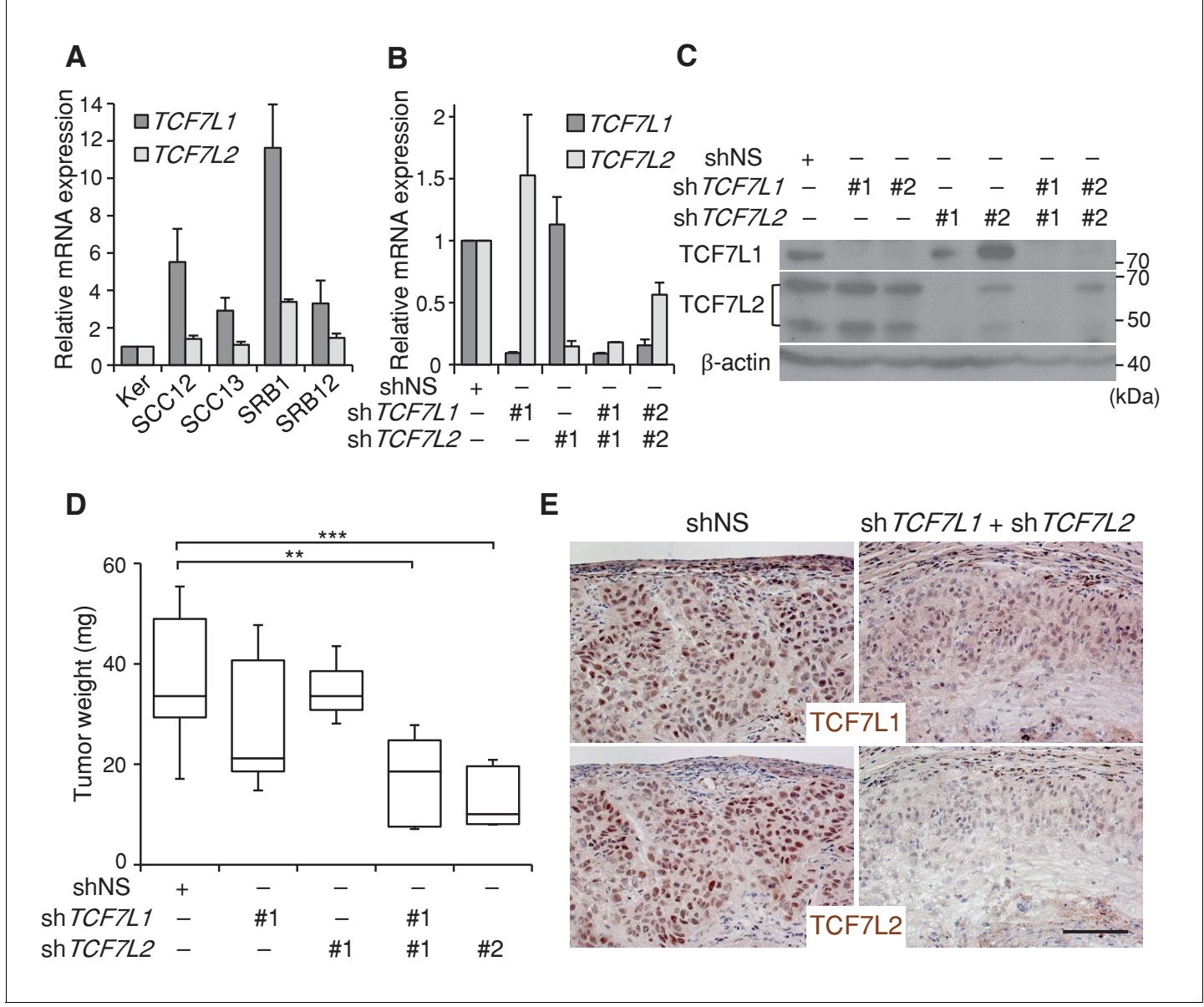

**Figure 3.** Downregulation of *TCF7L1* and *TCF7L2* reduces human skin SCC growth. (**A**) Expression of *TCF7L1* and *TCF7L2* in adult human keratinocytes (KER) and different human SCC cell lines, including SCC12, SCC13, SRB1 and SRB12. The human skin SCC cell line SCC13 was transduced to express tet-inducible control shRNA (shNS) or shRNAs against *TCF7L1* and/or *TCF7L2*. After drug selection, transduced cells were treated with doxycycline for five days prior to mRNA and protein isolation for (**B**) Real time PCR analysis or (**C**) Western analysis. The transduced cells with effective shRNAs were xenografted into flanks of immunodeficient mice, which were then put on a doxycycline-containing diet. Tumors were isolated and analyzed at the end of eight weeks. (**D**) Mass of tumors derived from SCC cells expressing control shRNAs (shNS), shRNAs against *TCF7L1* (sh*TCF7L1*#1), shRNAs against *TCF7L2* (sh*TCF7L2*#1), or both (sh*TCF7L1*#1+sh*TCF7L2*#1 or sh*TCF7L1*#2+sh*TCF7L2*#2). n = 19 (shNS), n = 5 (sh*TCF7L1*#1), n = 3 (sh*TCF7L2*#1), n = 5 (sh*TCF7L1*#1+sh*TCF7L2*#1), n = 5 (sh*TCF7L1*#2+sh*TCF7L2*#2). The grafted tumor mass data are presented as box and whisker plots where boxes span first and third quartiles, bars as the median values, and whiskers as minimum and maximum of all data. **$p<0.01$, ***$p<0.001$ (One-way ANOVA with Dunnett's post-hoc test). (**E**) Immunohistochemical analysis of TCF7L1 and TCF7L2 expression in xenografted tumors derived from SCC cells expressing control shRNAs (shNS) or shRNAs against both *TCF7L1* and *TCF7L2* (sh*TCF7L1*#1+sh*TCF7L2*#1). Bar denotes 100 μm.

The following figure supplement is available for figure 3:

**Figure supplement 1.** Silencing *TCF7L1* and *TCF7L2* in SCC cells reduces proliferation in vitro and in vivo.

against nonspecific sequence (shNS), sh*TCF7L1* or sh*TCF7L2* or both. The transduced cells were then grafted into nude mice, which were put on a doxycycline-containing diet the following day to attain expression of the specified shRNAs. We found that knocking down *TCF7L1* only modestly reduced tumor growth while knocking down both *TCF7L1* and *TCF7L2* significantly reduced the tumor size (*Figure 3D*). We confirmed by immunohistochemical analysis of the tumors that the induced shRNAs were efficient in reducing the expression of both TCF7L1 and TCF7L2 (*Figure 3E*). Consistent with the observation that silencing *TCF7L1* and *TCF7L2* reduced tumor size, their down-regulation reduced SCC cell proliferation in vitro and in vivo (*Figure 3—figure supplement 1*).

## Overexpression of TCF7L1 increases tumorigenic potential of human SCC cells

To assess whether TCF7L1 overexpression in human SCC cells drives tumor growth, we next evaluated the effect of TCF7L1 overexpression in the xenograft model of human skin SCC. We transduced SCC cell lines, SRB12 (*Rodríguez-Villanueva and McDonnell, 1995*) and SCC13 (*Rheinwald and Beckett, 1981*), with a lentiviral vector expressing tet-inducible myc-tagged *Tcf7l1* or empty vector as a control. We grafted the drug-selected transduced cells onto immunodeficient mice and placed the mice on a doxycycline-containing diet the following day. At the end point of the experiment, we measured the average mass of TCF7L1-overexpressing tumors and found it was significantly greater than that of the control tumors. TCF7L1 overexpression increased the weight of xenografted tumors derived from both skin SCC cell lines, SRB12 (*Figure 4A*) and SCC13 (*Figure 5D*).

We also tested the tumorigenic ability of TCF7L1-overexpressing SCC cells by grafting transduced cells at limiting dilution onto immunodeficient mice. We found that TCF7L1-overexpressing cells form tumors at a higher efficiency (*Figure 4B*). Immunofluorescence analysis of xenografted tumors showed that tumors expressing induced myc-TCF7L1 display increased proliferation, evident by the higher level of BrdU incorporation (*Figure 4C*, upper panel) and increased number of Ki67-positive cells (*Figure 4C*, lower panel, and *Figure 4D*).

## TCF7L1 promotes tumor growth independently of its binding to β-catenin

Since TCF7L1 plays independent roles in transcriptional regulation through binding to the transactivator *β*-catenin as well as the corepressor Groucho/TLE, we sought to determine which function of TCF7L1 is important for its tumorigenic activity. We compared the tumorigenic potential of full-length TCF7L1 to three mutant versions: TCF7L1ΔN, which lacks the *β*-catenin binding domain, TCF7L1ΔG, which lacks the Groucho/TLE corepressor binding domain, and TCF7L1*, which contains a mutation in its HMG domain and cannot bind to DNA (*Merrill et al., 2001*) (*Figure 5A*). As previously described (*Merrill et al., 2001*), TCF7L1ΔN lacked activating activity on WNT/*β*-catenin responsive promoter while TCF7L1ΔG showed robust activity (*Figure 5—figure supplement 1*). We transduced SCC13 cells with a lentiviral vector expressing GFP alone or with each form of the tet-inducible myc-tagged *Tcf7l1* mutants (*Miao et al., 2014*). After enriching the transduced cells by FACS, we verified by immunofluorescence and Western blotting analyses that TCF7L1 and its mutant versions were similarly expressed after doxycycline treatment (*Figure 5B,C*).

We injected transduced cells into nude mice and placed the grafted mice on a doxycycline-containing diet. At the end of eight weeks, we compared the mass of tumors derived from TCF7L1-overexpressing cells to those derived from control cells. We found that full-length TCF7L1 overexpression significantly increased tumor size relative to the control while TCF7L1*, which cannot bind to DNA, had little effect (*Figure 5D*). Notably, the *β*-catenin binding-deficient TCF7L1ΔN mutant increased tumor size similarly to full-length TCF7L1, whereas the Groucho/TLE corepressor binding-deficient TCF7L1ΔG did not significantly increase the tumor size (despite having a 100 fold stronger activity on WNT reporter in vitro). These data indicate that TCF7L1 does not require binding to *β*-catenin to promote human skin SCC tumor growth and suggest that TCF7L1 might act as a co-repressor in its tumor-promoting role.

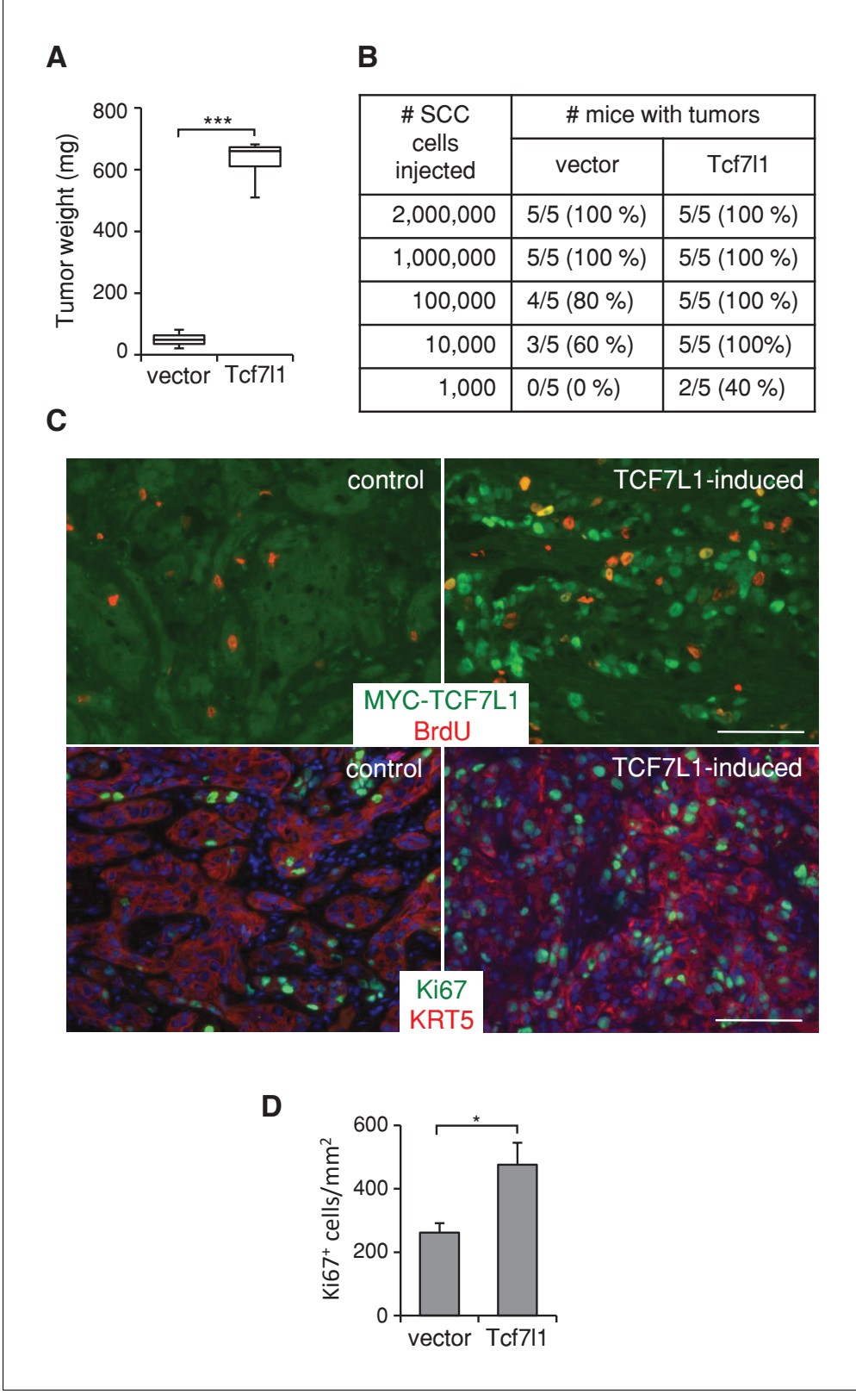

**Figure 4.** TCF7L1 stimulates tumor growth and increases tumorigenic capacity. (**A**) TCF7L1 promotes xenogafted tumor growth. Human skin SCC cell line SRB12 transduced with control vector or vector expressing tet-inducible *Tcf7l1* were drug selected and expanded. 1 million cells were xenografted onto NSG mice. Grafted mice were put on doxycycline-containing chow the next day. Tumor weight was measured at the end of 10 weeks. The grafted *Figure 4 continued on next page*

*Figure 4 continued*

tumor mass data are presented as box and whisker plots where boxes span first and third quartiles, bars as the median values, and whiskers as minimum and maximum of all data. n = 5. \*\*\*p<0.001 (Student T test). (B) Limiting-dilution transplantation assay from SRB12 cells with or without TCF7L1 overexpression. Transduced SRB12 cells were grafted onto NSG mice at serial diluted cell numbers as indicated. Mice were put on doxycycline-containing chow the next day and were given a pulse of BrdU 3 hr prior to tumor isolation at the end of 10 weeks. A presence of a tumor is scored when it is over 10 mg. (C) Immunofluorescence analysis of expression of induced TCF7L1 and proliferative markers BrdU (top panel) and Ki67 (bottom panel) with Keratin 5 marking epithelial cells. Bar denotes 100 μm. (D) Quantification of Ki67 positive cells in a minimum of 5 fields per tumor and three tumors/group. \*p<0.05 (Student T test).

## TCF7L1 overexpression overrides HRAS-induced senescence independently of β-catenin interaction

To gain insight into the mechanism underlying TCF7L1's ability to increase tumor incidence and progression in the mouse model of skin SCC, we examined TCF7L1's effect on oncogene-induced senescence (OIS), as senescence represents an early tumor suppressive barrier that must be overcome for tumor development to occur (*Collado and Serrano, 2010*). We transduced cells to express oncogenic HRAS$^{G12V}$ in the presence or absence of tet-inducible *TCF7L1*, and then evaluated the degree of cellular senescence by measuring senescence-associated β-galactosidase (SA-β-gal) activity. As expected, expression of oncogenic HRAS$^{G12V}$-induced senescence in both primary mouse keratinocytes (*Figure 6A–C*) and fibroblasts (*Figure 6—figure supplement 1*). However, when co-expressed with TCF7L1, its effect on senescence in keratinocytes was significantly reduced (*Figure 6A–C*). Importantly, TCF7L1 counteracted HRAS$^{G12V}$-induced senescence independently of its binding to β-catenin, as TCF7L1ΔN mutant exerted even greater negating effect than full-length TCF7L1 (*Figure 6D*). Consistent with our finding that TCF7L1 suppressed OIS independently of β-catenin, we found that OIS was not affected by the overexpression of ΔNβ-catenin, the N-terminus truncated version of β-catenin that is hyper-active due to its increased stability and consequent nuclear localization (*Gat et al., 1998*) (*Figure 6E*, *Figure 6—figure supplement 2*). We confirmed that ΔNβ-catenin expression was enriched in nuclei of transduced cells as expected (*Figure 6—figure supplement 2*). In fact, introducing ΔNβ-catenin together with TCF7L1 reduced the suppressing effect of TCF7L1 on OIS. Together, these data demonstrate that TCF7L1 acts independently of ΔNβ-catenin to suppress HRAS$^{G12V}$-induced senescence.

## TCF7L1 overexpression stimulates SCC cell migration independently of β-catenin interaction

In our previous study, we found that TCF7L1 accelerates migration of murine primary keratinocytes independently of β-catenin interaction (*Miao et al., 2014*). Using the Boyden chamber assay, we confirmed that TCF7L1 also stimulates cell migration in two different human SCC cell lines, SCC13 and SRB12 (*Figure 7*). We observed a significant increase in motility in cells expressing full-length TCF7L1 and TCF7L1ΔN, with little change upon TCF7L1ΔG expression. Thus, as with its role in promoting tumor growth and suppressing oncogene-induced senescence, TCF7L1 stimulated migration of skin SCC cells independently of its interaction with β-catenin.

## Overexpression of TCF7L2 promotes tumor growth and alters cell behavior similarly as overexpression of TCF7L1

Since downregulation of both *TCF7L1* and *TCF7L2* in SCC cells were required to reduce xenografted tumor growth, it suggested that TCF7L1 and TCF7L2 can compensate for each other and that they may have a similar function in skin SCC. Indeed, TCF7L2 and TCF7L1 showed similar activity on TCF/β-catenin responsive promoter in vitro (*Figure 5—figure supplement 1*). Importantly, overexpression of TCF7L2 also increased xenografted SCC tumor size, accelerated cell migration, and counteracted HRAS$^{G12V}$-induced senescence (*Figure 8*) similarly as TCF7L1.

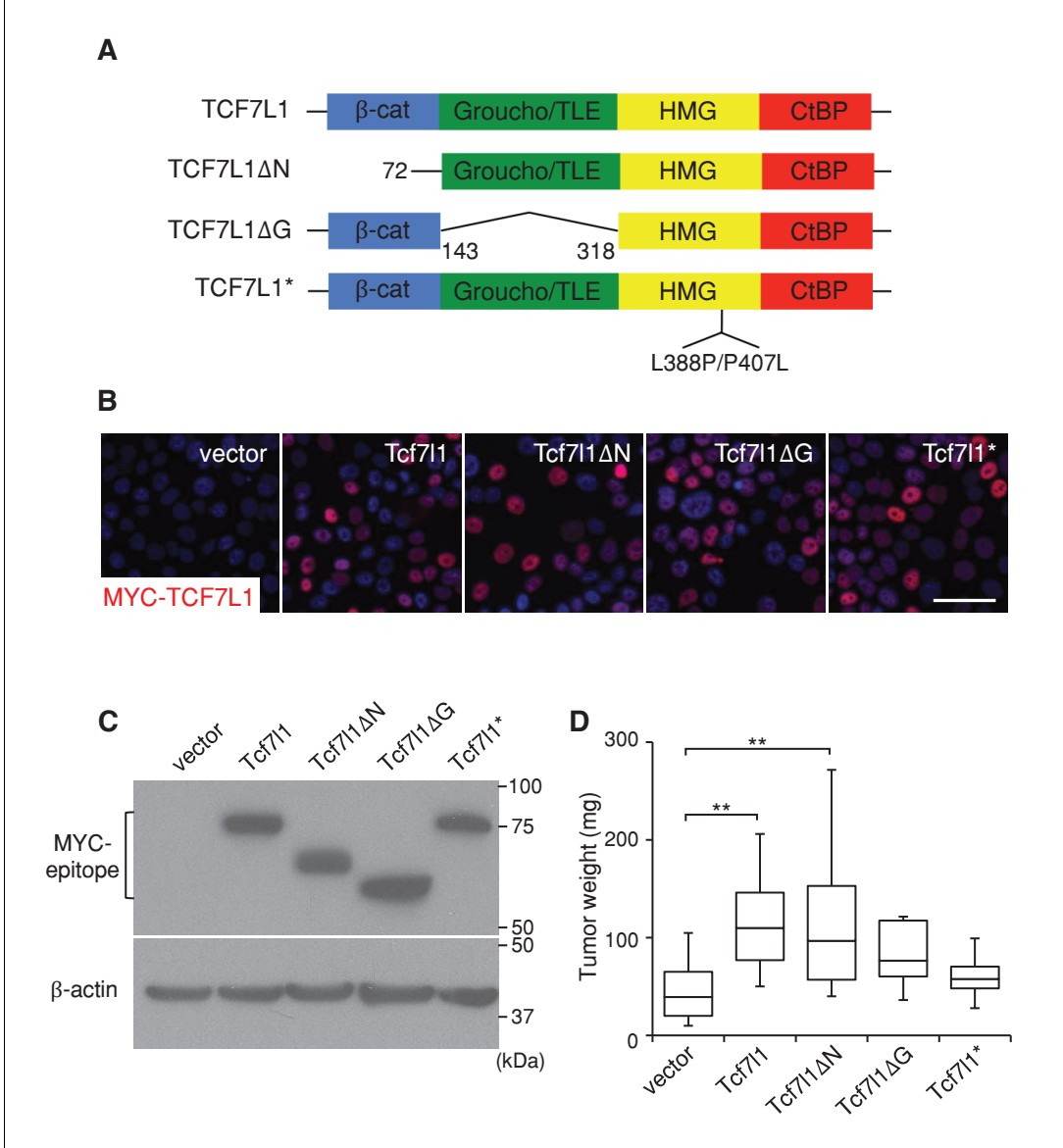

**Figure 5.** TCF7L1 promotes tumor growth independently of $\beta$-catenin binding. The human skin SCC cell line SCC13 was transduced with pINDUCER21 lentiviral vectors expressing GFP alone or GFP with tet-inducible, myc-tagged murine *Tcf7l1* or *Tcf7l1* deletion mutants. After enrichment by FACS, the transduced cells were xenografted into flanks of immunodeficient mice, which were then put on a doxycycline-containing diet. (**A**) TCF7L1 and its mutant versions are schematized with amino acid deletion and point mutation annotated. (**B**) Immunofluorescence analysis of myc-TCF7L1 expression in cells overexpressing TCF7L1 or its mutants. Bar denotes 50 µm. (**C**) Western analysis of protein expression against myc epitope in transduced cells. (**D**) Graph quantifying the mass of tumors derived from SCC cells with vector or the overexpression of *Tcf7l1* or its mutants. n = 12 (vector), n = 19 (Tcf7l1), n = 15 (Tcf7l1ΔN), n = 7 (Tcf7l1ΔG), n = 4 (Tcf7l1*). The grafted tumor mass data is presented as box and whisker plots where boxes span first and third quartiles, bars as the median values, and whiskers as minimum and maximum of all data **p<0.01 (One-way ANOVA with Dunnett's post-hoc test).

The following figure supplement is available for figure 5:

**Figure supplement 1.** Activity of TCF7L1, TCF7L1 deletion mutants and TCF7L2 on TCF/$\beta$-catenin responsive promoter (TOPFlash).

## TCF7L1 alters the transcriptomic landscape and promotes tumor growth through induction of LCN2

Since both TCF7L1 and TCF7L1ΔN increased tumor growth, suppressed senescence, and stimulated cell migration much more significantly than TCF7L1ΔG and TCF7L1*, we sought to identify the

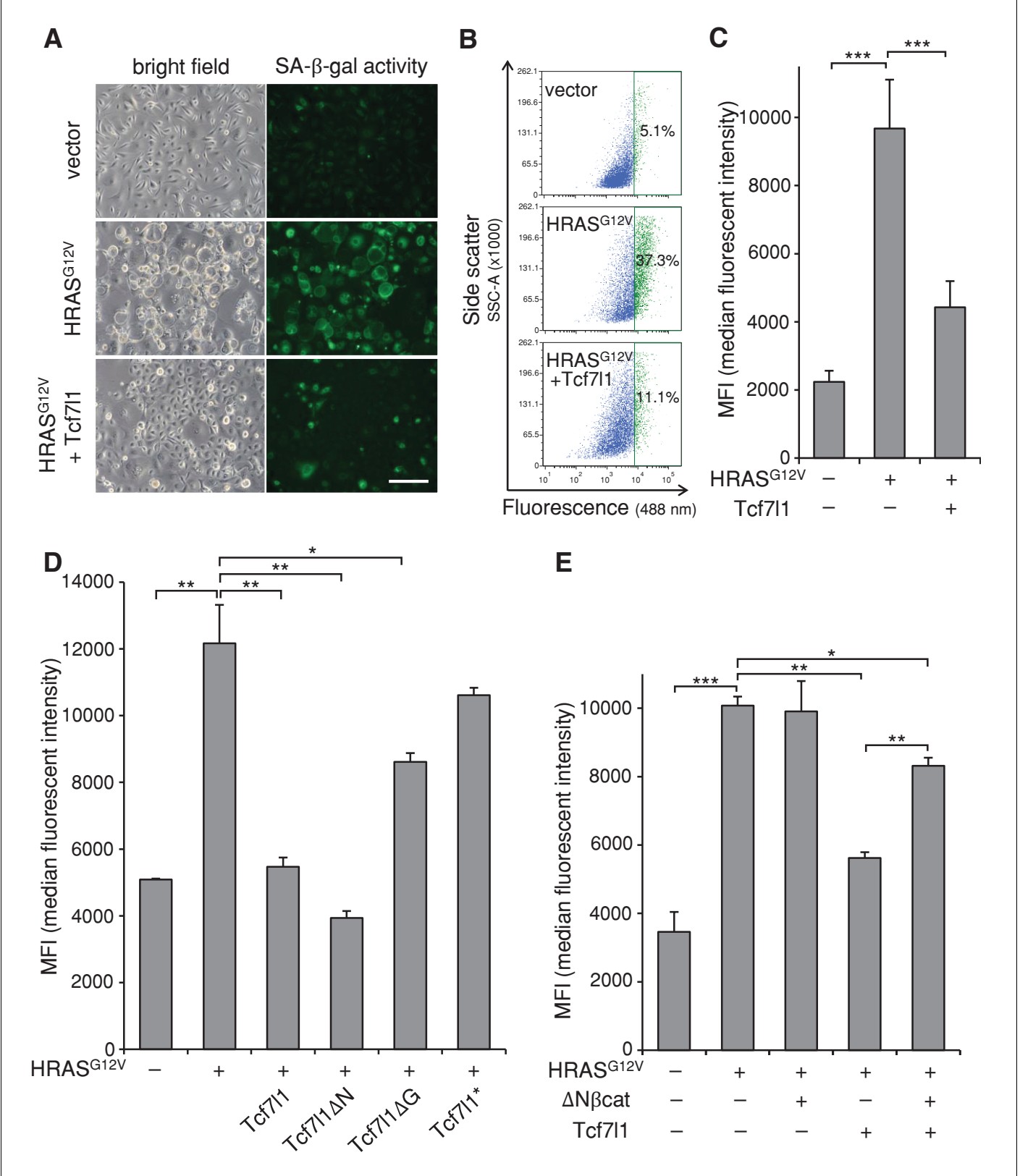

**Figure 6.** TCF7L1 suppresses HRAS[G12V]-induced senescence independently of *β*-catenin interaction. Mouse primary keratinocytes were transduced to express tet-inducible *Tcf7l1* or *Tcf7l1* deletion mutants. After cell line establishment by G418 selection, cells were re-infected to express oncogenic *HRAS*[G12V] or pBabe-puro vector as control. Cells were selected with puromycin, treated with doxycycline two days post-infection, and analyzed 5–6

*Figure 6 continued on next page*

*Figure 6 continued*

days post-infection as indicated. (A) Fluorescent detection of SA-$\beta$-gal activity five days post *HRAS^{G12V}* infection. Transduced cells were incubated with $\beta$-gal substrate $C_{12}$FDG (5-Dodecanoylaminofluorescein Di-$\beta$-D-Galactopyranoside), which is cleaved by $\beta$-gal to release a fluorescent product that is retained by the cells. Bar denotes 200 µm. (B) Flow cytometry analysis of SA-$\beta$-gal activity in mouse keratinocytes with or without *Tcf7l1* overexpression six days post *HRAS^{G12V}* infection. pBabe-transduced cells with $C_{12}$FDG incubation serve as control. Cells were pretreated with 300 µM chloroquine for lysosomal neutralization followed by $C_{12}$FDG substrate incubation for another 2 hr. Samples were then analyzed with Arial II flow cytometer using a 488 nm filter. Results were represented in dot plot (x axis, $C_{12}$FDG fluorescent intensity; y axis, side scatter in area) after size exclusion. The threshold to define SA-$\beta$-gal positive cells is set by 5% gate in the control sample as presented in green population. (C) SA-$\beta$-gal activity of each sample in B is indicated by median fluorescent intensity (MFI) after size exclusion. (D) Flow cytometry analysis of SA-$\beta$-gal activity in *HRAS^{G12V}*-transduced mouse keratinocytes with the overexpression of *Tcf7l1* or *Tcf7l1* mutants. (E) Flow cytometry analysis of SA-$\beta$-gal activity in *HRAS^{G12V}*-transduced mouse keratinocytes with the overexpression of $\Delta$N-$\beta$cat, *Tcf7l1*, or both. Data is presented as mean ± s.d. *p<0.05, **p<0.01, ***p<0.001 (One way ANOVA with Dunnett's post-hoc test).

The following figure supplements are available for figure 6:

**Figure supplement 1.** TCF7L1 overrides HRAS^{G12V}-induced senescence in fibroblasts.

**Figure supplement 2.** $\Delta$N$\beta$cat has no effect on HRAS^{G12V}-induced senescence in keratinocytes.

global change in gene expression caused by the overexpression of TCF7L1 and TCF7L1$\Delta$N and less so by the overexpression of TCF7L1$\Delta$G and TCF7L1*. We performed RNAseq analysis of SCC13 cells that were transduced and drug selected to express vector control, full-length TCF7L1 or its different variants, using the Illumina NextSeq 500 system to yield an average of 17 million reads. Gene expression heatmaps of these transduced cells illustrate that indeed TCF7L1 and TCF7L1$\Delta$N share a similar altered gene signature. The difference in gene expression pattern between the control and the TCF7L1 and TCF7L1$\Delta$N group is more pronounced than its difference with the TCF7L1$\Delta$G and TCF7L1* group (*Figure 9A*).

Since overexpression of TCF7L1* had little effect on tumor size, OIS, and migration while TCF7L1$\Delta$G still affected OIS and SCC cell migration, albeit significantly less than the full length and TCF7L1$\Delta$N (*Figures 5D*, *6D* and *7B*), we decided to focus on the genes that were altered by TCF7L1 and TCF7L1$\Delta$N but not by TCF7L1*. Venn diagrams constructed with gene sets that were significantly altered (Q < 0.05) in three groups identified 359 genes that were induced ($\geq$2 fold) and 43 genes that were repressed ($\geq$2 fold) by both TCF7L1 and TCF7L1$\Delta$N but not by TCF7L1* (*Figure 9B*). We further performed GO term and KEGG pathway analyses of these induced and repressed specified gene sets (*Figure 9C*; Supplementary Data 1). Among the listed GO biological processes (p<0.05), the top three induced GO terms were inflammation, response to lipopolysaccharide, and response to molecule of bacterial origin. Among the listed KEGG pathways (p<0.05), the top three induced GO terms were genes involved in complement and coagulation cascades, cytokine-cytokine receptor interaction, and TNF signaling pathway. Inflammation-related factors such as *LTA*, *RRAD*, *LCN2* and *TNF* were included in the top 20 significantly induced gene list (Q < 0.05) (*Figure 9D,E*).

Of the top genes altered by TCF7L1 and TCF7L1$\Delta$N and not TCF7L1*, we were drawn especially to *LCN2*, since we previously identified LCN2 as a downstream effector of TCF7L1 that stimulates epidermal cell migration (*Miao et al., 2014*). In addition, because TCF7L1 overexpression seems to have a paracrine effect on proliferation in vivo (*Figure 2—figure supplement 1*, *Figure 4C*) and that LCN2 is a secreted protein overexpressed in many types of cancer (*Li and Chan, 2011*), we surmise that TCF7L1 may drive tumor growth through induction of LCN2.

In both SCC13 and SRB12 human SCC cell lines, we verified by real time PCR that overexpression of either TCF7L1 or TCF7L1$\Delta$N increased expression of *LCN2* more significantly than the overexpression of TCF7L1$\Delta$G, while overexpression of TCF7L1* had little effect (*Figure 10A*). We showed that overexpression of TCF7L1 induced expression of LCN2 in vivo (*Figure 10—figure supplement 1A*) as previously reported (*Miao et al., 2014*) and that expression of LCN2 was also induced in murine skin papilloma and SCC (*Figure 10C*, *Figure 10—figure supplement 1B*) similarly to TCF7L1.

We next sought to determine whether expression of TCF7L1 is correlated with LCN2 expression in human skin SCC. Through immunostaining of human skin samples (n = 16), we found that while 10

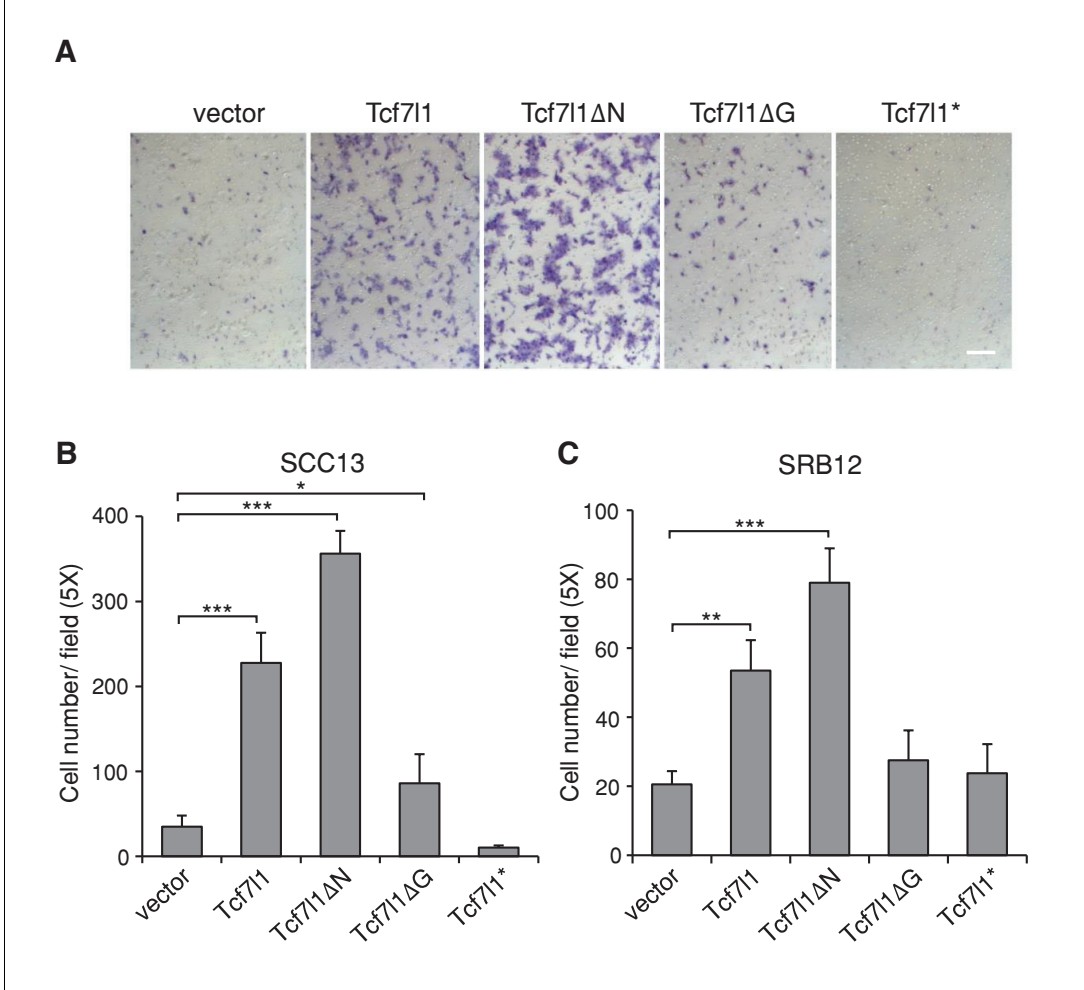

**Figure 7.** TCF7L1 promotes cell migration in SCC cells independently of *β*-catenin binding. The human skin SCC cell lines SCC13 and SRB12 were transduced with tet-inducible *Tcf7l1* or *Tcf7l1* deletion mutants. After drug selection, the transduced cells were treated with doxycycline for 48 hr and their migratory ability was measured using the Boyden chamber-based cell migration assay. Bar denotes 200 μm. (A) Representative images of migrated SCC13 cells stained with crystal violet. (B–C) Quantification of migrated SCC cells, SCC13 (B) or SRB12 (C), per 5X field. Data are presented as mean ± s.d. *p<0.05, **p<0.01, ***p<0.001 (One-way ANOVA with Dunnett's post-hoc test).

out of 16 samples were positive for TCF7L1, 15 were positive for LCN2 (*Figure 10—figure supplement 2*). In addition, although all of TCF7L1 expressing tumors express LCN2, 83.3% of TCF7L1 negative also express LCN2, indicating that LCN2 can be induced by factors other than TCF7L1.

To test whether LCN2 is a major downstream effector of TCF7L1 that stimulates tumor growth, we took a combined gain- and loss-of-function strategy. After validating that two different sets of shRNAs against *LCN2* can efficiently downregulate the expression levels of both mRNA and protein (*Figure 10C,D*), we transduced SRB12 cells to express vector control or tet-inducible *Tcf7l1* with shRNAs against nonspecific sequence (shNS) or *LCN2* (sh*LCN2*#1 and sh*LCN2*#2). We grafted the transduced cells onto NSG mice and placed these grafted mice on a doxycycline-containing diet the following day. After eight weeks, TCF7L1-overexpressing tumors were significantly bigger than the control tumors. Importantly, downregulating *LCN2* decreased tumor size, abrogating the effect of TCF7L1 overexpression on tumor growth (*Figure 10E,F*). Moreover, as was similarly shown in primary murine keratinocytes (*Miao et al., 2014*), inhibition of LCN2 with neutralizing antibody abolished the promigratory effect of TCF7L1 in SCC cells (*Figure 10G*).

Since LCN2 has been shown to stimulate neutrophil chemotaxis in vitro (*Shao et al., 2016*), and inflammation facilitates tumor growth, we evaluated whether overexpressing TCF7L1 increases

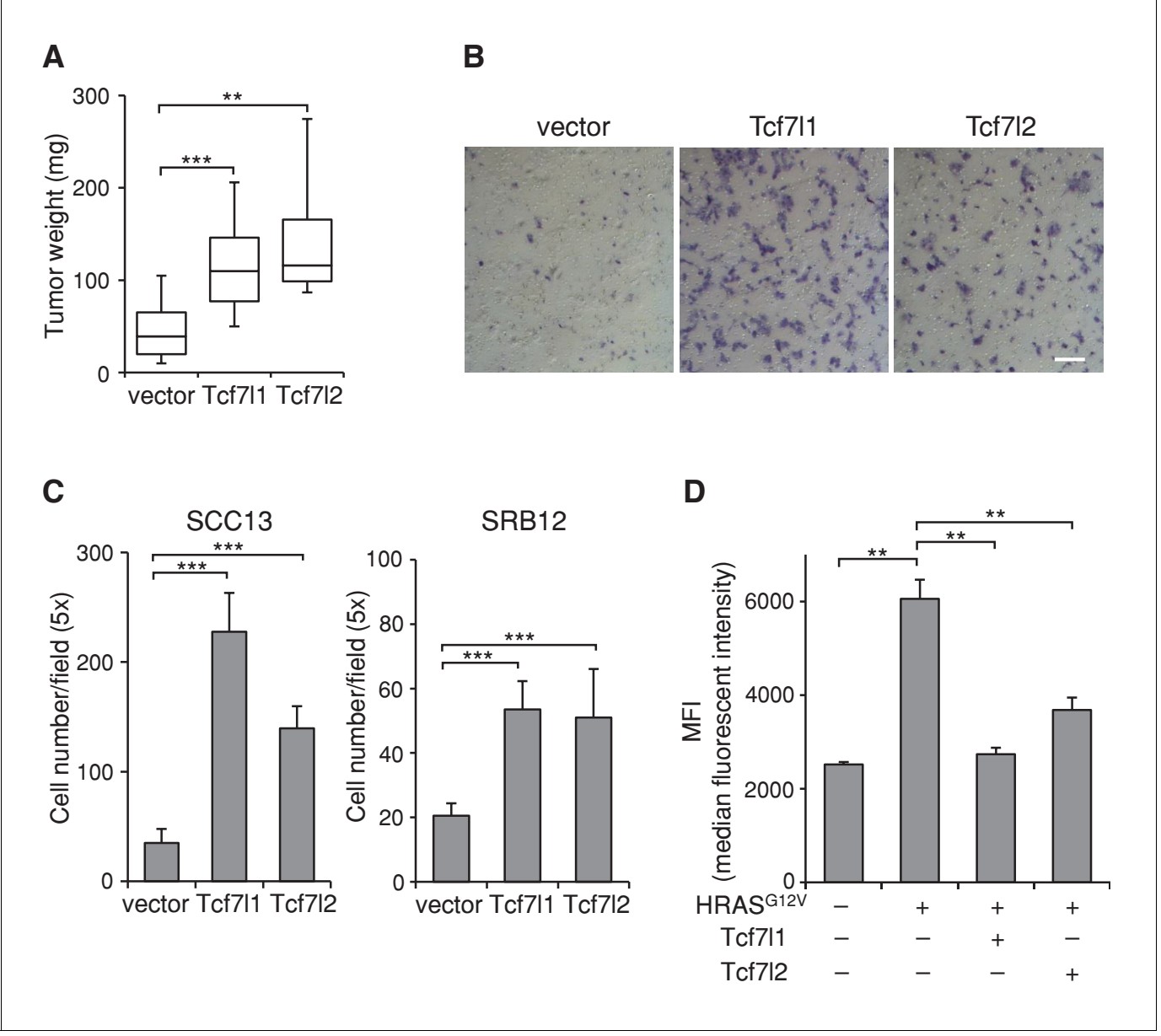

**Figure 8.** Overexpression of TCF7L2 promotes tumor growth, increases cell migration, and represses HRAS[G12V]-induced senescence similarly to TCF7L1. The human skin SCC cell line SCC13 was transduced to express tet-inducible, myc-tagged TCF7L1 or TCF7L2. After G418 drug selection, transduced cells were grafted into flanks of immunodeficient nude mice, which were then put on a doxycycline containing diet for eight weeks. (**A**) Graph quantifying the mass of tumors derived from SCC cells with vector or the overexpression of *Tcf7l1* or *Tcf7l2*. n = 11 (vector), n = 18 (Tcf7l1), n = 4 (Tcf7l2). The grafted tumor mass data is presented as box and whisker plots where boxes span first and third quartiles, bars as the median values, and whiskers as minimum and maximum of all data. **p<0.01, ***p<0.001 (One-way ANOVA with Dunnett's post-hoc test). (**B–C**) Boyden chamber-based cell migration assay. SCC cells were treated with doxycycline for 48 hr to overexpress TCF7L1 or TCF7L2. (**B**) Representative image of migrated SCC13 cells stained with crystal violet. Bar denotes 200 μm. (**C**) Quantification of migrated SCC13 or SRB12 SCC cells per 5X field. Data are presented as mean ± s.d. ***p<0.001 (One-way ANOVA with Dunnett's post-hoc test). (**D**) Flow cytometry analysis of SA-β-gal activity in *HRAS[G12V]*-transduced mouse keratinocytes with the overexpression of TCF7L1 or TCF7L2. The SA-β-gal activity of each sample is indicated by median fluorescent intensity (MFI) with 5000 cells (after size exclusion) per sample in duplicates. Data are presented as mean ± s.d. **p<0.01 (One-way ANOVA). Note that the overexpression of TCF7L2 experiments were performed together with the overexpression of TCF7L1 experiments, whose data were shown in *Figures 5–7*.

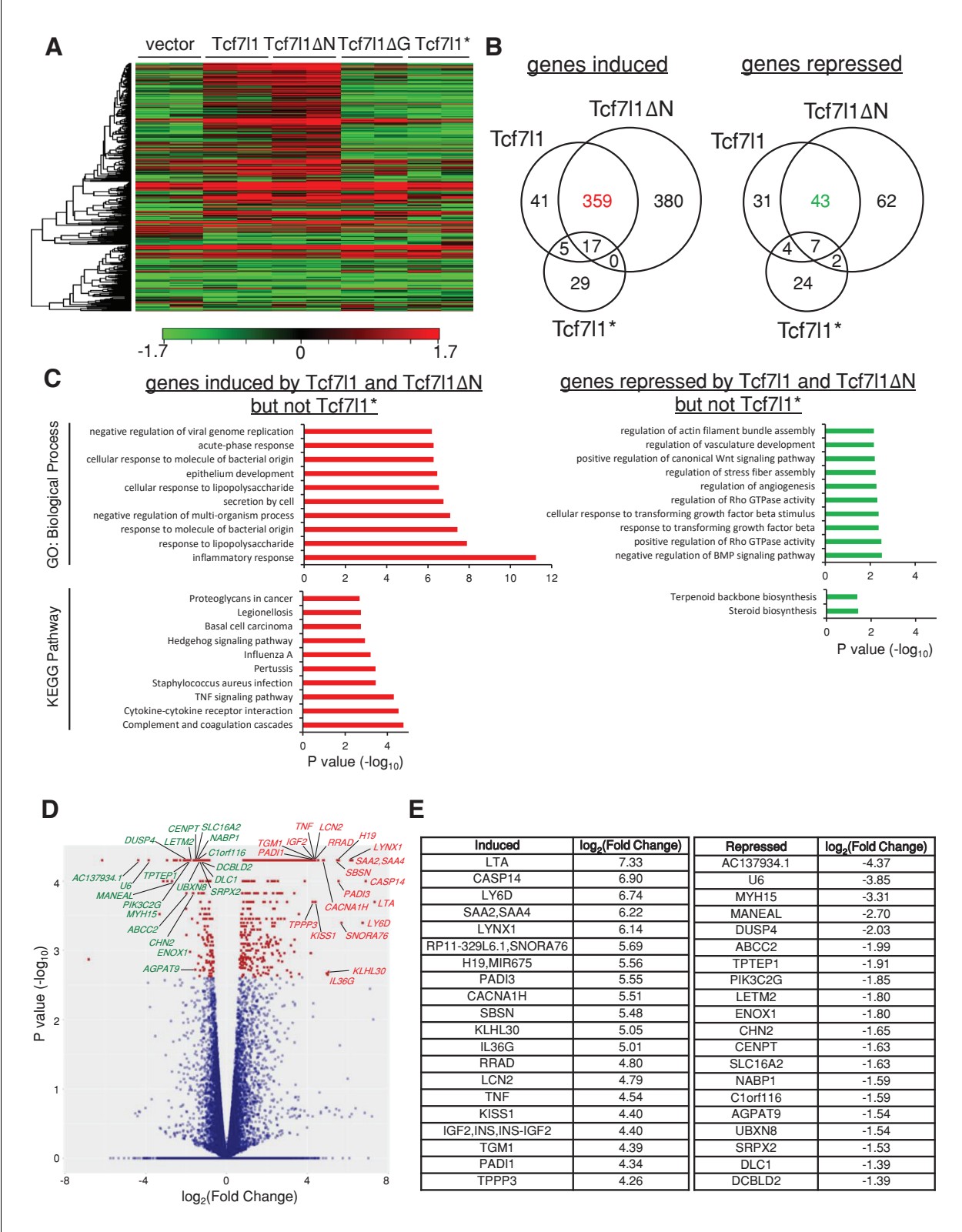

**Figure 9.** TCF7L1 overexpression alters transcriptional landscape in SCC cells. The human skin SCC cell line SCC13 was transduced with lentiviral vectors expressing tet-inducible *Tcf7l1* or *Tcf7l1* deletion mutants. After drug selection, the transduced cells were treated with doxycycline for 48 hr and RNA was harvested for gene expression profiling by RNAseq. (**A**) Heat map of genes differentially expressed in SCC13 cells with vector, overexpression of *Tcf7l1* or *Tcf7l1* deletion mutants. Gene list is ordered by clustering. Induced gene sets are shown in red, and repressed gene sets are shown in

*Figure 9 continued on next page*

Figure 9 continued

green. (B) Venn diagram of genes induced (as indicated in red) or repressed (as indicated in green) by both TCF7L1 and TCF7L1ΔN but not TCF7L1*. Numbers of overlaid genes were annotated across groups. Differentially expressed genes from each group were those with significant difference (q < 0.05) above the threshold, $\log_2$(Fold Change)>=1 or =<−1. (C) Biological process and KEGG pathway analyses of differentially expressed genes sets in B. Gene set analysis (GSA) was performed using Enrichr tool (*Chen et al., 2013*). The list includes up to 10 significantly involved terminologies (p<0.05) for GO ontology (biological process) and Kyoto Encyclopedia of Genes and Genomes (KEGG) pathways. List of top 20 genes significantly induced or repressed by TCF7L1 and TCF7L1ΔN but not TCF7L1* in (D) Volcano plot and in (E) Table with corresponding fold change. Outliers were excluded from the list.

neutrophil infiltration in tumors and whether downregulating LCN2 alters their infiltration. Using flow cytometry analysis, we quantified the number of neutrophils (CD11b+Ly6G+Lyc6C+) and macrophages (CD11b +F4/80+) in xenografted tumors expressing control vector or *Tcf7l1* with or without downregulation of *LCN2* (*Figure 10—figure supplement 3*). We found that while overexpression of TCF7L1 did not affect the number of macrophages (vector: 2.78 ± 1.57% vs. TCF7L1: 2.27 ± 1.02%), it increased the number of neutrophils by four fold (vector: 0.36 ± 0.09% vs. TCF7L1: 2.28 ± 0.74%). However, downregulation of LCN2 did not impinge on the ability of TCF7L1 to increase neutrophil infiltration. Together, our data therefore imply that TCF7L1 accelerates human SCC cell migration and tumor growth through induction of LCN2 but does not require LCN2 to stimulate neutrophil infiltration.

## Discussion

*TCF7L1* is an embryonic stem cell signature gene that is upregulated in multiple aggressive cancer types, including breast cancer, glioblastoma, and bladder carcinoma (*Ben-Porath et al., 2008*). In a xenograft model of breast cancer, downregulation of *TCF7L1* decreased tumor growth and reduced metastasis rate (*Slyper et al., 2012*). Because downregulating *TCF7L1* in these cells resulted in simultaneous upregulation and downregulation of different subsets of WNT target genes (*Slyper et al., 2012*), it was thought that TCF7L1's function as a β-catenin binding transcriptional coactivator might explain its role in promoting tumorigenesis. This hypothesis was supported by the finding that β-catenin is essential for the development and growth of papillomas in a murine skin SCC model, and that *Tcf7l1* mRNA is upregulated in papilloma (*Malanchi et al., 2008*). However, whether the overexpression of TCF7L1 contributes to the development of premalignant tumors and SCC in skin and whether it requires β-catenin interaction in this function remain to be elucidated.

Here we report that TCF7L1 protein is upregulated in premalignant and malignant tumors in both the chemically- and UV-induced mouse models of skin SCC (*Figure 1*). Using the tet-inducible *Tcf7l1* transgenic mouse model to mimic the upregulation of TCF7L1 found in papilloma and skin SCC, we demonstrated that TCF7L1 overexpression promotes and accelerates both premalignant papilloma formation and progression into SCC (*Figure 2E–G*).

In the DMBA/TPA chemical carcinogenesis protocol, because the tumor-initiating step (DMBA) is distinct from the promoting step (TPA), we could identify which phase of tumorigenesis was impacted by TCF7L1 overexpression by omitting each agent. Although TCF7L1 overexpression was not sufficient to promote tumor development without DMBA/TPA (*Figure 2—figure supplement 3*), the overexpression of TCF7L1 partially compensated for TPA administration in promoting tumor development, suggesting that TCF7L1 acts as a tumor promoter (*Figure 2H–J*). Since prolonged overexpression of TCF7L1 increased proliferation in vivo (*Figure 2—figure supplement 1*), it is reasonable that TCF7L1 overexpression can partially fill the role of TPA, which is known to induce cell proliferation (*Yuspa et al., 1982*; *Hennings et al., 1987*) and expansion of mutated clones (*Karen et al., 1999*). However, it should be noted that while overexpression of TCF7L1 increased the incidence of papilloma development in DMBA/TPA treated mice, it did not increase the average size of papilloma (data not shown). Since overexpression of TCF7L1 did not stimulate proliferation of DMBA/TPA induced papilloma per se and did not promote tumor development without DMBA treatment (*Figure 2—figure supplement 3C*), it suggests that TCF7L1 may facilitate the transitioning of DMBA-induced mutated cells into premalignant tumors.

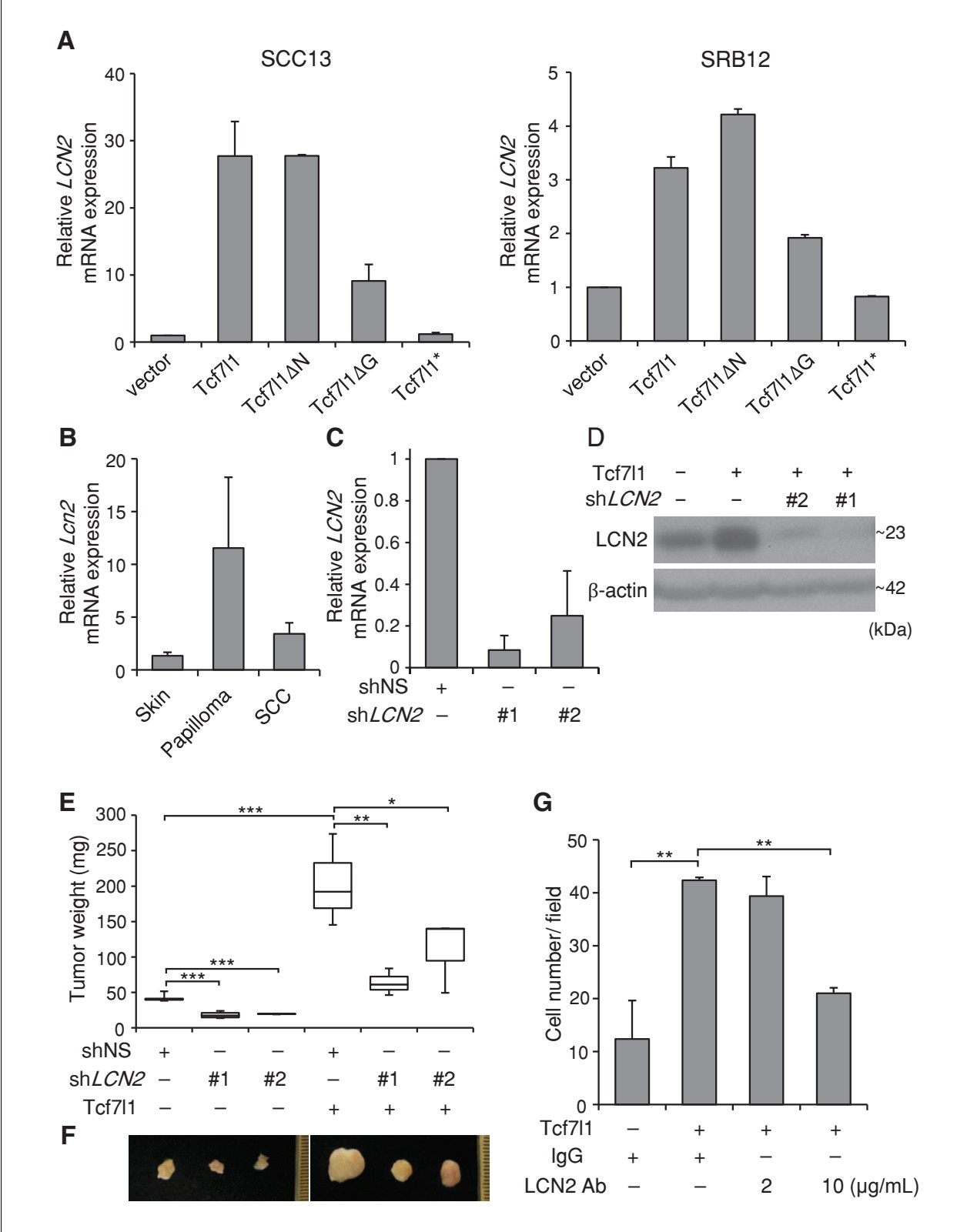

**Figure 10.** TCF7L1 promotes tumor growth through induction of *LCN2*. (**A**) Real time PCR analysis of expression of *LCN2* in SCC13 and SRB12 cells that were transduced to express *Tcf7l1* and its various mutants. (**B**) Real time PCR analysis of *Lcn2* in enriched epithelial cells from mouse normal skin, DMBA/TPA-induced papilloma and SCC. (**C**) Real time PCR analysis of *LCN2* in SRB12 cells that were transduced to express shRNA against nonspecific sequence (shNS) or two different sets of shRNAs against *LCN2*. (**D**) Western analysis of LCN2 in SRB12 cells that were transduced to express empty

*Figure 10 continued on next page*

*Figure 10 continued*

vector control or *Tcf7l1* with shRNA against nonspecific sequence or two different sets of shRNAs against *LCN2.* Drug selected transduced cells expressing shRNAs were grafted onto NSG mice and tumors were isolated at the end of eight weeks. (**E**) Quantification of tumor mass. Data are presented as box and whisker plots where boxes span first and third quartiles, bars as the median values, and whiskers as minimum and maximum of all data. *p<0.05, **p<0.01, ***p<0.001 (One-way ANOVA with Dunnett's post-hoc test). n = 5 (shNS+vector), n = 4 (sh*LCN2*#1+vector), n = 2 (sh*LCN2*#2 +vector), n = 3 (shNS+*Tcf7l1*), n = 3 (sh*LCN2*#1+*Tcf7l1*), n = 3 (sh*LCN2*#2+ *Tcf7l1*). (**F**) Representative images of tumors. (**G**) Quantification of cell migration in transwell assay. Human SCC (SRB12) cells with tet-inducible *Tcf7l1* were treated with doxycycline for 48 hr and their migratory ability was measured using the Boyden chamber-based cell migration assay. Cells were allowed to migrate toward the lower compartment containing media with 10% FBS as chemoattractant, supplemented with neutralizing antibody against LCN2 or mouse IgG isotype control. After 30 hr, migrated cells were fixed by 4%PFA, counterstained with crystal violet, and quantified. Data are presented as mean ± s.d. **p<0.01 (One-way ANOVA with Dunnett's post-hoc test).

The following figure supplements are available for figure 10:

**Figure supplement 1.** LCN2 is induced by TCF7L1 and is overexpressed in murine skin papilloma and SCC.

**Figure supplement 2.** TCF7L1 and LCN2 are expressed in human skin SCC.

**Figure supplement 3.** Overexpression of TCF7L1 increases neutrophil infiltration independently of LCN2.

In a xenograft model of human skin SCC, downregulation of *TCF7L1* and its paralogue *TCF7L2* decreased tumor size, indicating that these two LEF/TCF family members, whose functions overlap in skin development (*Nguyen et al., 2006*), are required for the optimal growth of xenografted tumors (*Figure 3*). Conversely, overexpression of TCF7L1 increased tumorigenic capacity of SCC cells in limiting dilution assay and stimulated tumor growth (*Figure 4*). While overexpression of TCF7L1 stimulated SCC growth in vivo, it did not alter SCC cell proliferation in vitro (data not shown), suggesting that xenografted TCF7L1-overexpressing SCC cells acquire characteristics that confer growth advantage in vivo.

Our data clearly demonstrate that TCF7L1 exerts a tumorigenic function in both the chemical-induced mouse model of skin SCC and the xenograft model of human skin SCC. Our finding that TCF7L1 overexpressing mice developed papilloma at a higher number and at an earlier time under both DMBA/TPA or DMBA alone treatment indicates that TCF7L1 overexpression contributes to the early step of tumor formation. Since we saw a higher number of TCF7L1-overexpressing mice developed SCC, we infer that TCF7L1 overexpression also influences malignant progression. Our finding that TCF7L1-overexpressing human SCC cells formed larger xenografted tumors and at a higher efficiency further corroborates that TCF7L1 contributes to the malignant nature of SCC cells.

These data together suggest that elevated expression of TCF7L1 facilitates the transitioning into the premalignant state as well as the progression from the premalignant state to SCC. This model for TCF7L1's role in the development and progression of skin SCC was also supported by the observation made by Chitsazzadeh et al. that expression of the target genes of TCF7L1 are significantly altered not only between normal skin and papilloma/actinic keratosis samples, but also between papilloma/actinic keratosis samples and skin SCC in both human and mice (*Chitsazzadeh et al., 2016*).

Since $\beta$-catenin is critical for tumor growth in the DMBA/TPA model of skin SCC, and both nuclear $\beta$-catenin and *Tcf7l1* mRNA are upregulated in papillomas (*Malanchi et al., 2008*), it was thought that TCF7L1's tumor-promoting role might depend on its binding to $\beta$-catenin in the canonical WNT signaling pathway. However, our data demonstrate that TCF7L1 promotes tumor growth independently of its interaction with $\beta$-catenin, requiring only its binding to DNA and Groucho/TLE corepressors (*Figure 5*). Our results imply that TCF7L1 might act as a transcriptional repressor in its tumor-promoting role, further supporting the model that TCF7L1 functions primarily as a repressor and not as a WNT target gene-activating $\beta$-catenin cofactor as shown in a variety of developmental processes (*Wray et al., 2011*; *Yi et al., 2011*; *Kim et al., 2000*; *Pereira et al., 2006*; *Gribble et al., 2009*; *Wu et al. 2012*; *Kuwahara et al., 2014*; *Miao et al., 2014*). However, we should note that the deleted region in TCF7L1ΔG encompassing the Groucho/TLE binding domain is quite large and may contain binding interfaces to other proteins. Therefore, our interpretation should be viewed

with the caveat that TCF7L1ΔG may be deficient in binding to other factors than just to Groucho/TLE.

TCF7L1 has been shown to promote tumor growth in both xenograft models of human breast cancer (*Slyper et al., 2012*) and colorectal cancer (*Murphy et al., 2016*), but it is not clear whether TCF7L1 functions as an activator or repressor to exert its tumorigenic role. In colorectal cancer cell line HCT116, downregulation of TCF7L1 increased WNT reporter TOP-FLASH activity in vitro, suggesting that TCF7L1 functions as a repressor. In breast cancer cell line MDA-MB-468, overexpression of TCF7L1 together with β-catenin increased TOP-FLASH activity, suggesting that it functions as a coactivator with β-catenin. Moreover, since downregulation of TCF7L1 in breast cancer cells decreased the expression of some WNT-responsive genes while increasing a different subset, it implies that TCF7L1 can act as an activator or repressor depending on the targets. However, to date no experiments have been conducted in vivo to identify which function of TCF7L1 is required for tumorigenesis. Therefore, although that TCF7L1 may function as an activator and repressor in breast cancer cells, its tumorigenic role may depend only on its function as a repressor. Our work is the first to use deletion mutations of TCF7L1 to show that TCF7L1 does not require binding to β-catenin to promote tumor growth. It is possible that this repressor function of TCF7L1 may be similarly required in cancers other than skin SCC, but that remains to be determined experimentally in vivo. What is consistent in all three tissue models (breast, colorectal, and skin) is that TCF7L1 plays a pro-tumorigenic role.

Since TCF7L1 overexpression promoted and accelerated skin tumor development, we explored possible mechanisms underlying this role. Previous studies have shown that mutation of the *Ras* oncogene is an early event in the DMBA/TPA carcinogenesis model. As early as 3–4 weeks after DMBA treatment, activating mutations in *Hras1* can be detected in the epidermis before any development of papillomas (*Nelson et al., 1992*). Subsequently, these mutations are found in the majority of papillomas that are induced by DMBA/TPA (*Balmain et al., 1984*). We hypothesized that TCF7L1 overexpression may help tumor cells overcome oncogene-induced senescence, which is a phenomenon observed both in vitro (*Serrano et al., 1997*) and in vivo (*Braig et al., 2005*; *Chen et al., 2005*; *Collado et al., 2005*; *Lazzerini Denchi et al., 2005*; *Michaloglou et al., 2005*). In tissues where expression of oncogenic HRAS promotes tumor formation, senescence markers are found abundantly in premalignant lesions but are absent in carcinomas (*Collado et al., 2005*). These findings led to the model that overcoming senescence is a crucial step in malignant progression. Indeed, in a separate study, premalignant tumors induced by oncogenic HRAS overexpression progressed into malignancy only after the senescence pathway was abrogated by genetic ablation of the tumor suppressor gene *Trp53* (*Sarkisian et al., 2007*).

We found that TCF7L1 overexpression overrides oncogenic HRAS-induced senescence (*Figure 6*). Although these in vitro results suggest that this may be a mechanism by which TCF7L1 overexpression increases tumor incidence and growth, we currently lack the in vivo data to support this claim due to technical limitations associated with detecting senescent markers in mouse epidermis. Detection of SA-β-gal has been reported in aged human epidermis (*Dimri et al., 1995*), but its staining has been negative in aged mouse epidermis (*Gannon et al., 2011*) unless senescence is activated by genetic manipulation of senescence genes (*Tokarsky-Amiel et al., 2013*). We were unable to detect SA-β-gal activity in aged mouse epidermis or in tested skin papillomas (data not shown). Similarly, by immunostaining for expression of p16[INK4a] as well as LaminB, whose respective increased and decreased levels have been used to show senescence in aged human skins, we failed to see expected changes in aged mouse epidermis. Indeed, although these markers have been used to differentiate young and aged human skin, they have so far failed to be validated in aged mouse skin by immunostaining (*Gannon et al., 2011*; *Tokarsky-Amiel et al., 2013*). Due to this technical limitation to detect senescence in mouse skin in vivo, our finding regarding TCF7L1 overriding senescence is based only on in vitro experimental data. Nevertheless, we could conclude that at least in vitro TCF7L1 is able to override HRAS-induced senescence mainly through its DNA- and Groucho/TLE corepressor binding domains and not on its β-catenin binding domain, similarly to its ability to promote tumor growth.

Besides overcoming HRAS-induced senescence, prolonged TCF7L1 overexpression also stimulated proliferation of both murine interfollicular epidermis and hair follicles in vivo (*Figure 2—figure supplement 1*). It has been shown that quiescent hair follicle stem cells are less prone to cancer initiation in response to activating *Kras* mutations (*Kras^{G12D}*) and p53 ablation (*White et al., 2014*). It is

thus possible that TCF7L1 overexpression increases tumor incidence and/or progression by promoting the proliferation of quiescent stem cells with oncogenic mutations. Since non-TCF7L1 overexpressing epidermal cells in the TCF7L1-overexpressing epidermis (due to mosaicism) also showed increased proliferation (*Figure 2—figure supplement 1*, *Figure 4C*), we surmise that overexpression of TCF7L1 might affect proliferation in a paracrine manner. However, paracrine effect of TCF7L1 on proliferation remains to be tested, as we lack an in vitro assay to evaluate the effect of conditioned-media from TCF7L1-overexpressing cells, given that TCF7L1 overexpressing cells have similar proliferative capacity in vitro.

Although TCF7L1-overexpressing skins are hyperproliferative, control and TCF7L1-induced papilloma do not differ in size. Since overexpression of TCF7L1 increased papilloma incidence but not papilloma size, we postulate that TCF7L1 facilitates the transformation of mutated cells into premalignant state and contributes little to growth of papilloma. Moreover, since TCF7L1-induced mice developed more SCCs, we speculate that overexpression of TCF7L1 also stimulates the progression of papilloma into malignancy.

We should point out that in contrast to the DMBA/TPA model where TCF7L1 overexpression increased proliferation of epidermal cells in skins that have yet to become papilloma but not in the papilloma themselves, in a xenograft model overexpression of TCF7L1 increased proliferation of SCC cells in vivo (*Figure 4C,D*). We found that TCF7L1 overexpression also induced cell migration in human SCC cell lines (*Figure 7*), as was observed in mouse primary keratinocytes (*Miao et al., 2014*). Similar to its ability to increase tumor growth and suppress oncogene-induced senescence, its promigratory function requires binding to DNA and Groucho/TLE and not to $\beta$-catenin.

Our transcriptome analyses of SCC cells expressing TCF7L1 or its mutant variants found that overexpression of TCF7L1 or TCF7L1ΔN transforms the global gene expression profile to a similar extent (*Figure 9A*, *Supplementary file 1*). This is not surprising given that overexpression of TCF7L1 and TCF7L1ΔN produced a similar effect on tumor growth, OIS, and cell migration. Since cells overexpressing TCF7L1 and TCF7L1ΔN showed the highest tumorigenic activity while TCF7L1*-overexpressing cells were indistinguishable from control cells, we chose to focus on genes that were altered by TCF7L1 and TCF7L1ΔN but not TCF7L1*. Using this metric, we identified a specified induced and repressed set of genes controlled by TCF7L1 that may be important to the growth of skin SCC. We should note that since our profiling studies were performed on SCC13 cells, which are already malignant, our study is likely to identify only TCF7L1-responsive genes that contribute to increased SCC growth and may miss out the downstream effectors of TCF7L1 that facilitate the early step of tumor formation.

Further GO analysis of specified genes altered by TCF7L1 and TCF7L1ΔN in SCC13 cells identified the inflammatory response as the top altered biological process. Induced genes include TNF secreted factors such as *LTA* (160.9 fold), *LTB* (8.2 fold), *TNF* (23.3 fold), *TNFSF8* (2.6 fold), *TNFSF10* (4.2 fold), *TNFSF14* (3.0 fold), and *TNFSF18* (3.1 fold), pro-inflammatory factors such as *IL6* (4.6 fold), *IL8* (2.9 fold), *LCN2* (27.7 fold) and *MMP9* (13.3 fold) and a subset of TNF receptors, including *TNFRSF1B* (13.7 fold), *TNFRSF8* (3.4 fold), *TNFRSF9* (3.6 fold), *NGFR* (5.5 fold), and *TNFRSF18* (4.1 fold). TNF and IL6 have been well defined as inflammatory cytokines in cancer-associated inflammation (*Suganuma et al., 2002*). Mice deficient of TNF or IL6 are more resistant to DMBA/TPA-induced skin tumor formation (*Moore et al., 1999*; *Ancrile et al., 2007*). Our study discovered a causative link between TCF7L1 up-regulation and cancer-associated inflammation in skin tumorigenesis, suggesting that TCF7L1 may facilitate tumor growth through induction of a subset of cytokines in a paracrine manner.

Of the top altered inflammatory genes, we focused on *LCN2*, because we previously found that LCN2 acts as a major downstream effector of TCF7L1 that induces murine epidermal cell migration (*Miao et al., 2014*). The paracrine effect of TCF7L1 on cell migration in vitro and wound closure in vivo (*Miao et al., 2014*) suggested that the secreted LCN2 might be a prime effector of TCF7L1 in skin tumorigenesis. Consistent with its proposed role in skin SCC, expression of LCN2 is undetectable in normal epidermis and is robustly upregulated in both murine and human skin SCC (*Figure 10B*, *Figure 10—figure supplements 1* and *2*). While overexpression of TCF7L1 in skin induced expression of LCN2 (*Figure 10—figure supplement 1A*), it did not further increase the already upregulated level of LCN2 in the tumors (data not shown). Since LCN2 is upregulated in TCF7L1-induced skins but not upregulated any higher in TCF7L1-induced tumors, it suggests that factors regulating the expression of LCN2 including TCF7L1 may already be in excess in the tumors;

hence further increases of TCF7L1 do not affect expression of LCN2. Since overexpression of TCF7L1 predisposed DMBA/TPA treated skins to develop tumors at a higher rate but did not affect tumor size, it suggests that upregulation of LCN2 by TCF7L1 in the pre-tumor state may be facilitating the transitioning of mutated cells into tumor cells.

In our xenograft model, we found that downregulating LCN2 in human SCC cells counteracted the pro-tumorigenic and promigratory effect of TCF7L1 (*Figure 10E–G*), demonstrating that LCN2 acts a major downstream effector of TCF7L1 in these cells. Although LCN2 has been reported to stimulate chemotaxis of neutrophils in vitro (*Shao et al., 2016*), we found that downregulating LCN2 did not diminish the ability of TCF7L1 to stimulate neutrophil infiltration in tumors (*Figure 10—figure supplement 3*). Together, the data show that TCF7L1 requires LCN2 to induce growth and migration but works independently of LCN2 to stimulate neutrophil infiltration.

Apart from its role in inducing cell migration, which other cellular processes LCN2 acts on to stimulate tumor growth remains to be elucidated. Moreover, how overexpression of TCF7L1 leads to increased transcription of *LCN2* remains to be elucidated. Since TCF7L1-induced *LCN2* expression depends mainly on DNA interaction but not on $\beta$-catenin interaction (*Figure 10A*), we suspect that TCF7L1 acts on yet to be identified direct targets that consequently bring about the induction of *LCN2*. Given that LCN2 is induced in the majority of human skin SCC, and is expressed in both TCF7L1-positive and -negative samples (*Figure 10—figure supplement 2*), factors other than TCF7L1 must also upregulate LCN2 in tumors. Indeed, NF-κB (*Shen et al., 2006*; *Li et al. 2009*; *Karlsen et al., 2010*; *Mahadevan et al., 2011*), C/EBP (*Shen et al., 2006*; *Glaros et al., 2012*), and NFAT1 (*Gaudineau et al., 2012*) all have been shown to activate expression of LCN2 in response to various stress and injury stimuli.

In summary, we established that TCF7L1 promotes the development of papilloma and skin SCC, affecting both the early step of tumor formation as well as the progression step into malignancy in the DMBA/TPA model. Using a xenograft model of human skin SCC, we demonstrated that TCF7L1 stimulates tumor growth, suppresses oncogene-induced senescence, and accelerates tumor cell migration, independently of its interaction with $\beta$-catenin. Although we do not yet know the direct repressed target genes of TCF7L1 that contribute to skin tumorigenesis, we clearly showed that LCN2 is a major downstream effector of TCF7L1 that drives tumor growth. Our findings thus provide mechanistic insight into a transcription factor that is upregulated in multiple aggressive cancer types and plays an important role in skin tumorigenesis.

## Materials and methods

### Mice breeding and carcinogenesis protocol

*KRT14-rtTA* transgenic mice (RRID:IMSR_JAX:008099) were crossed with *TRE-mycTcf7l1* mice to generate the double transgenic line in FVB/N background as described (*Nguyen et al., 2006*). In the DMBA/TPA carcinogenesis model, mice of specified genotypes were put on doxycycline (200 mg/kg) containing chow (BioServ, Flemington, NJ) at eight weeks of age. One week later, the mice were subjected to the established 2-stage DMBA/TPA protocol (*Abel et al., 2009*), at either a high dose of DMBA (100 nmol) and TPA (10 nmol) or a low dose of DMBA (25 nmol) and TPA (1 nmol) in 200 µL of acetone per mouse. Briefly, dorsal skin was painted with a single dose of 7,12-Dimethylbenz[a]anthracene (DMBA) (Sigma-Aldrich, St. Louis, MO), followed by twice-weekly topical application of Phorbol 12-myristate 13-acetate (TPA) (Sigma-Aldrich) for 25 weeks. For experiments in which DMBA or TPA was omitted, 200 µL of acetone vehicle control was applied on each mouse instead. Mice were sacrificed 25 weeks post DMBA treatment or when the total tumor burden reaches 1.5 cm as allowed by IACUC.

In the UV-induced carcinogenesis model, 90 day old SKH1-E hairless female mice (RRID:IMSR_CRL:477) were irradiated with 12.5 kJ/m$^2$ of UVB weekly (ILT1700/73 for 100 days using solar simulators (*Benavides et al., 2009*) (Oriel, Newport, CA). Normal unirradiated skin (abdomen), papillomas, and SCC were harvested 14 days after the last day of irradiation. Papillomas developed within about 100 days following initiation of exposure and about 15% of these lesions evolved into invasive SCC (*Vin et al., 2013*).

All mice were maintained in the AALAC-accredited animal facilities at Baylor College of Medicine and MD Anderson and all mouse experiments were conducted according to protocols approved by committees at Baylor College of Medicine (AN-4907) and MD Anderson (ACUF00001396-RN00).

## Xenograft

Human SCC cell lines SCC13 (*Rheinwald and Beckett, 1981*) or SRB12 (*Rodríguez-Villanueva and McDonnell, 1995*) were transduced to express genes or shRNAs against genes of interest, as specified in each experiment. Transduced cells were FACS sorted or drug selected and expanded in vitro for grafting. For SCC13 xenograft, $2 \times 10^6$ cells in serum-free basal medium with an equal volume of growth factor-reduced, phenol red free Matrigel (BD, ThermoFisher, Waltham, MA) were injected in a total volume of 200 μL into each flank of female athymic nude mice (CrTac:NCr-Foxn1$^{nu}$ from Taconic RRID:IMSR_TAC:ncrnu) at 8 to 12 week-old. For SRB12 xenograft, $2 \times 10^6$ cells in 200 μL serum-free basal medium were injected into each flank of NSG mice (NOD *scid* gamma, NOD-*scid* IL2Rg$^{null}$, NOD-*scid* IL2gamma$^{null}$ from Jackson Lab RRID:IMSR_JAX:005557). Grafted mice were given doxycycline-containing chow (200 mg/kg) (BioServ) the following day to induce tet-inducible gene or shRNA expression. Tumors were harvested and analyzed at 8 week post transplantation. All xenograft experiments were performed with gender-matched and age-matched (at age difference within 10 days) mice in the same batch. Tumors were measured by tumor mass (mg). For tumorigenicity assay, serial dilution of SRB12 cells were prepared and transplanted into the flank of the NSG mice. After 10 weeks, the presence of tumor outgrowth was evaluated to estimate the tumor formation frequency from a given cell population. The presence of a tumor is scored when the mass of the harvested tumor is over 10 mg.

## Cell culture and cell line establishment

Mouse primary keratinocytes were isolated from newborn pups and cultured as described (*Miao et al., 2014*). Human SCC cell lines SCC12 (RRID:CVCL_4026) and SCC13 (RRID:CVCL_4029 ) were kindly provided by Dr. James Rheinwald and have been previously characterized (*Rheinwald and Beckett, 1981*). SRB1 (RRID:CVCL_AT72) and SRB12 (RRID:CVCL_AT73) have been characterized (*Rodríguez-Villanueva and McDonnell, 1995*) and provided by Dr. Kenneth Tsai (M. D. Anderson, now at Moffit). These human SCC cell lines are not on the list of the commonly misidentified cell lines as established by the International Cell Line Authentication Committee and were authenticated by STR profiling. Human primary keratinocytes (ZenBio, Research Triangle Park, NC) and human SCC cell lines SCC12 and SCC13 were grown in Gibco K-SFM serum-free media (Life Technologies, Carlsbad, CA). Human SCC cell lines SRB1 and SRB12 were maintained in HyClone DMEM/F12 media (with 2.5 mM L-Glutamine and 15 mM HEPES Buffer) (VWR, Sugarland, TX) with 10% FBS, 1X Pen/Strep (Life Technologies), 100 mg/L Normocin (InvivoGen, San Diego, CA). All cells were mycoplasma tested and maintained in a humidified incubator with 5% $CO_2$ at 37°C.

## Plasmid construction

Tet-inducible-*Tcf7l1* and its mutants in lentiviral vector were generated as described (*Miao et al., 2014*). Myc tagged *Tcf7l1* and its mutants, including *Tcf7l1ΔN*, *Tcf7l1ΔG* and *Tcf7l1\**, in pENTR1A entry vector were subcloned to pINDUCER21 or pINDUCER20 vectors (*Meerbrey et al., 2011*) after gateway recombination. pLenti-RFP-*ΔNβ*-catenin and pLenti-RFP were generated by gateway recombination. pLenti-RFP(DEST) vector was modified from pLenti 7.3 vector (Life Technologies) by replacing GFP with turoboRFP fragment. pENTR1A entry vector with *β-globin* intron or with *KRT14-ΔN87β-cat*-derived (*Gat et al., 1998*) *ΔNβ-catenin* fragment were gateway recombined with pLenti-RFP(DEST) vector to generate pLenti-RFP or pLenti-RFP-*ΔNβ*-catenin respectively. All clones were verified by sequencing. shRNA constructs against *TCF7L1* or *TCF7L2* were generated by oligo ligation to Tet-pLKO-puro (Addgene plasmid # 21915) or Tet-pLKO-neo (Addgene plasmid # 21916) vectors respectively. shRNA constructs against *LCN2* were generated by oligo ligation to Tet-pLKO-puro. All clones were verified by sequencing. Target sequences are listed below. sh*TCF7L1*#1 (CCAGCACACTTGTCTAATAAA) (TRCN0000021705), sh*TCF7L1*#2 (TGAAGGAAAGTGCAGCCATTA) (TRCN0000021707), sh*TCF7L2*#1 (TAGCTGAGTGCACGTTGAAAG) (TRCN0000262843), sh*TCF7L2*#2 (GTCGACTTCTTCCTTACATTC) (TRCN0000262849), sh*LCN2*#1 (GGAGCTGAC

TTCGGAACTAAA) (TRCN0000372827), sh*LCN2*#2 (GTACTTCAAGATCACCCTCTA) (TRCN0000060290) and shNS (CAACAAGATGAAGAGCACCAA).

## Retroviral and lentiviral infection

Lentivirus production and infection were performed as described (*Miao et al., 2014*). Retrovirus production and infection were performed similarly except without the co-transfection of helper plasmids and virus being harvested at 32°C instead of 37°C. In brief, 293 T cells were transfected at 70–80% confluency using TransIT-293 transfection reagent (Mirus, Madison, WI). Lentiviral vectors were co-transfected with helper plasmids to generate viral supernatants at 37°C, followed by filtration through 0.45 mm syringe filter. Mouse keratinocytes or SCC cells were seeded in 100,000 cells per 3.5 mm well and infected the next day with GFP-tagged virus at M.O.I=5, or with other virus in 1 mL supernatant per well. Viral infection was mediated by polybrene and enhanced by centrifugation at 1100 xg, 33°C, for 30 min. Stable cell lines were isolated by GFP-based fluorescence-activated cell sorting (FACS) (FACSAria II flow cytometer, BD) or by drug selection: 200 µg/mL G418 (InvivoGen, San Diego, CA) and/or 1.5 µg/mL puromycin (InvivoGen) for mouse keratinocytes or for SCC13 cells; 800 µg/mL G418 and/or 1.5 µg/mL puromycin for SRB12 cells.

## Real-time RT-PCR analysis

cDNA was synthesized from 500 ng RNA using SuperScript First-Strand Synthesis System (Life Technologies) and was diluted 20-fold for real-time PCR analysis. Real-time PCR was performed using Takyon No ROX SYBR Mastermix blue dTTP (AnaSpec, Fremont, CA) and the LightCycler 480 real-time PCR system (Roche, Indianapolis, IN). Primers for measuring relative gene expression were listed as follows. *TCF7L1* (forward, 5′-TATTTCGCCGAAGTGAGAAGGC-3′; reverse, 5′-TGACCTCG TGTCCTTGACTGT-3′), *TCF7L2* (forward, 5′-GCCTCTTATCACGTACAGCAAT-3′; reverse, 5′-GCCAGGCGATAGTGGGTAAT-3′), *LCN2* (forward, 5′-CCACCTCAGACCTGATCCCA-3′; reverse, 5′-CCCCTGGAATTGGTTGTCCTG-3′), *Lcn2* (forward, 5′-CCCTGTATGGAAGAACCAAGGA-3′; reverse, 5′-CACACTCACCACCCATTCAGT-3′).

## Gene expression profiling

Transduced and drug selected SCC13 cells expressing vector or tet-inducible *Tcf7l1* and its mutants were treated with 200 ng/mL doxycycline for 48 hr. For RNA extraction, cells were harvested into 1 mL Ribozol RNA extraction reagent (Amresco, Solon, OH), followed by RNA purification using PureLink RNA Mini Kit (Ambion, ThermoFisher). All samples were evaluated by 2100 Bioanalyzer (Agilent Genomics, Santa Clara, CA) with Agilent 6000 nano kit (Agilent Genomics) and samples with RNA integrity (RIN) >9 were proceeded with cDNA library prep. cDNA library was constructed by TruSeq stranded mRNA library prep kit LT (Illumina, San Diego, CA) according to manufacturer's protocol. After library prep, size distribution was determined by 2100 Bioanalyzer with Agilent DNA 1000 kit and the average sizes of the libraries were around 260 bp, as recommended by Illumina. In order to pool equal amount of libraries with different adaptors indexes, quantitative PCR were performed using Quantitative PCR KAPA Library Quantification Kit for Illumina (Kapa Biosystems, Wilmington, MA). All sequencing runs were performed on NextSeq 500 system (Illumina) using Next-Seq500 mid-output v2 paired-end sequencing kit, 150 cycles (FC-404–2001, Illumina). Differential gene expression and heatmap were generated by XploreRNA NGS data analysis (Exiqon, Denmark). In brief, RNA-seq analysis pipeline was based on the Tuxedo software package, including Bowtie2 (v2.2.2) for sequence alignment, TopHat (v2.0.11) for slice junction mapping and Cufflinks (v2.2.1) or sequence assembly.

## Western blotting

SCC cells and mouse keratinocytes that were transduced to express tet-inducible genes of interest were seeded in 6-well plates and were treated with 200 ng/mL doxycycline for 48 hr before harvesting. Cells were washed with PBS, lysed with 1x RIPA protein lysis buffer containing 1x Xpert Phosphatase inhibitor cocktail solution and 1x Xpert Protease inhibitor cocktail solution (GenDepot, Barker, TX), and sonicated for 15 s on ice. Cells lysates were cleared by centrifugation for 15 min at 13,500 xg at 4°C and the supernatant was collected. Protein concentration was quantified by Pierce BCA protein assay kit (ThermoFisher). 30 µg of protein was loaded to 10% SDS-PAGE, transferred

on nitrocellulose membrane and blotted for proteins of interest using antibodies as follows. Rabbit anti-Myc epitope (1:1,000, Cell Signaling Technology #2278 RRID:AB_490778), guinea pig anti-TCF7L1 (1:1,000, lab-generated), rabbit anti-TCF7L2 (1:1,000, Cell Signaling #2569S RRID:AB_2199816), rabbit anti-human LCN2 (1:1,000, Abcam #ab125075 RRID:AB_10978084), mouse anti-$\beta$-actin (1:5,000, Sigma #A5441 RRID:AB_476744). The following secondary antibodies from Jackson ImmunoResearch Labs (Westgrove, PA) were used: goat anti-rabbit IgG-HRP (#111-035-144 RRID: AB_2307391); goat anti-guinea pig IgG-HRP (#106-035-003 RRID:AB_2337402); and goat anti-mouse IgG-HRP (#115-035-003 RRID:AB_10015289). Image was developed with SuperSignal West Pico Chemiluminescent Substrate (ThermoFisher) and exposed to CL-XPosure X-ray film (ThermoFisher).

## Histological analyses

Mouse skin tissues were freshly-embedded and frozen in OCT compound or formalin fixed and embedded in paraffin. 5 µm paraffin sections or 8 µm OCT sections were used for analyses as described (*Miao et al., 2014*). 5 µm sections of de-identified paraffin-embedded human skin SCC samples (human exempt protocol H-25723) were obtained from the Pathology Cores at Baylor College of Medicine and Michael E. DeBakey VA Medical Center. For immunohistochemical analysis, microwave-based heat-induced epitope retrieval (HIER) was performed by boiling slides in 10 mM citrate buffer (pH 6.0) for 20 min, 5 min per round, four rounds in total. Endogenous peroxidase activity was then blocked in 3% hydrogen peroxide ($H_2O_2$) for 15 min at room temperature. Endogenous biotin was blocked by integrating an Avidin/Biotin blocking kit (Vector Laboratories, Burlingame, CA) with serum blockage and primary antibody incubation for overnight at 4°C. After Biotin-labeled secondary antibody incubation (1:200) and signal amplification by ABC kit (Vector Laboratories), color was developed with DAB chromogen (ImmPACT DAB; Vector Laboratories) and sections were counterstained with hematoxylin. For immunofluorescence analysis, OCT sections were fixed in 4% paraformaldehyde (PFA) for 10 min, PBS washed, and serum blocked (10% normal donkey serum, 2% BSA, 2% fish skin gelatin, and 2% Triton X-100 in PBS) for 1 hr at room temperature. Primary antibodies were used at the following concentration: guinea pig anti-TCF7L1 (1:200 for mouse and 1:100 for human; lab-generated), rabbit anti-TCF7L2 (1:200 for mouse and 1:100 for human; Cell Signaling #2569S RRID:AB_2199816), goat anti-mouse LCN2 (1:150, R and D Systems #AF1857 RRID:AB_355022); rabbit anti-human LCN2 (1:200, Abcam #ab125075 RRID:AB_10978084); rat anti-CD104 (1:200, BD Biosciences # 553745 RRID:AB_395027); rabbit anti-Myc epitope (1:200, Cell signaling #2278 RRID:AB_490778); rabbit anti-Ki67 (1:500, Leica Microsystems #NCL-Ki67p RRID:AB_442102); rat anti-BrdU (1:200, Abcam #ab6326, RRID:AB_305426); chick anti-Keratin 5 (1:500, BioLegend #905901 RRID:AB_2565054). Secondary antibodies from Jackson ImmunoResearch Labs were used as listed: anti rabbit FITC (1:150, #711-095-152 RRID:AB_2315776); anti-rat FITC (1:150, #712-095-153 RRID:AB_2340652); anti-goat FITC (1:150, #705-095-003 RRID:AB_2340400); anti-chicken RRX (1:200, #703-295-155 RRID:AB_2340371), anti-rabbit RRX (1:200, #711-295-152 RRID:AB_2340613); or anti-rabbit Alexa Fluor 647 (1:200, #711-605-152 RRID:AB_2492288) were used. Nuclei were counterstained with Hoechst 33342 (ThermoFisher). Images were acquired with a Zeiss Axiokop microscope.

## Flow cytometric measurement of senescence-associated $\beta$-galactosidase (SA-$\beta$-gal) activity

Mouse keratinocytes were transduced and drug selected to express empty vector or tet-inducible *Tcf7l1* and its mutants prior being infected with retrovirus carrying pBabe-puro-*HRAS^G12V* (Addgene plasmid # 9051) or pBabe-puro control vector (Addgene plasmid # 1764). Puromycin selection (1.5 µg/mL) and doxycycline (200 ng/mL) treatment began two days post infection. SA-$\beta$-gal activity was evaluated six days post *HRAS^G12V* infection. For determining the impact of stabilized $\beta$-catenin, cells were viral-infected by lentivirus harboring pLenti-RFP-$\Delta$N-$\beta$-catenin or pLenti-RFP empty vector two days post *HRAS^G12V* infection, followed by puromycin selection and doxycycline treatment. SA-$\beta$-gal activity was evaluated six days post *HRAS^G12V* infection. For microscopic imaging, mouse keratinocytes were incubated with 33 µM $\beta$-galactosidase substrate, 5-Dodecanoylaminofluorescein Di-$\beta$-D-Galactopyranoside ($C_{12}$FDG) (Setareh Biotech, Eugene, OR), for 2 hr at 37°C, and washed twice with PBS before being imaged with a 488 nm filter. For flow cytometry analysis, keratinocytes were pre-incubated with 300 µM chloroquine for 1 hr to neutralize internal pH of lysosomes to ~pH 6.0 before

substrate incubation in 33 µM $C_{12}$FDG and 100 µM chloroquine for 2 hr at 37°C. Cells were then PBS washed, trypsinized and strained through a 60 µm filter for flow analysis with a 488 nm filter. After size exclusion to remove cell debris, 5000 gated cells were analyzed per sample in duplicates. $C_{12}$FDGpositive cells were quantified based on the 5% threshold in the control sample (pBabe vector-transduced cells with $C_{12}$FDG incubation). Degree of senescence (SA-$\beta$-gal) was presented by median fluorescent intensity (MFI) of gated population. For cell population with pLenti-RFP-$\Delta$N$\beta$-catenin or pLenti-RFP vector, only RFP positive cells were gated for quantification and analyzed for $C_{12}$FDG signal intensity (MFI). FCS files were exported to be analyzed and graphed with FCS Express 6 Flow Cytometry (De Novo Software, Glendale, CA).

## Boyden chamber-based cell migration assay

SCC cells that were transduced and drug selected to express tet-inducible *Tcf7l1* and its mutants were treated with Doxycycline for two days prior to being trypsinized and used in cell migration assay. 100,000 SCC cells in 200 µL serum-free basal media with 0.5% BSA were seeded to a 24-well transwell insert with 8 µm pore size (BD, ThermoFisher), with 500 µL 10% FBS-containing DMEM/F12 media in the lower chamber. Cells were allowed to migrate for 26–30 hr, 4% PFA-fixed, and counterstained with crystal violet before imaging. At least four microscopic images per sample at 5X field were taken for quantifying migrated cells. To perform cell migration assay with neutralizing antibody against LCN2, 500 µL 10% FBS-containing DMEM/F12 media in the lower chamber was supplemented with LCN2 antibody (2 to 10 µg/mL, Abcam #ab23477 RRID:AB_447460) or mouse IgG isotype control (10 µg/mL, Jackson ImmunoResearch Labs #015-000-003 RRID:AB_2337188, ).

## MTT cell proliferation assay

SCC cells were seeded in 5000 cells/96-well and doxycycline-treated the next morning (day 0). MTT (3-(4,5-dimethylthiazol-2-yl)−2,5-diphenyltetrazolium bromide; Calbiochem, MilliporeSigma, Germany) reagent was added to 100 µL of media in final concentration as 0.5 mg/mL and incubated for 4 hr. After 15 min of DMSO incubation with gentle shaking in dark, cell growth was determined by microplate reader at 570 nm in absorbance. A growth curve was generated after seven days of measurement.

## TOPFlash reporter assay

Mouse keratinocytes transduced and drug selected to express tet-inducible *Tcf7l1* and its various mutants were seeded in 20,000 cells/24-well and transfection was performed the next day using TransIT-keratinocyte transfection reagent (Mirus Bio, Madison, WI) according to the manufacturer's instruction. In brief, transfection mixture was prepared by supplementing TransIT-keratinocyte (3 µL:1 µg) to serum free media with plasmid DNA including TOPFlash (Addgene plasmid # 12456), *KRT14* vector or *KRT14-ΔN87β-cat*, and *Renilla-TK* (internal control). After 20 min of incubation, transfection reagent was added to cells and media was then refreshed the next day with the addition of doxycycline to allow transgene induction. Samples were harvested by 1X passive protein lysis buffer (Promega, Madison, WI) 72 hr post transfection and stored at −20°C before luciferase measurement. Luciferase activity was determined using dual-luciferase reporter assay system (Promega).

## Measurement of senescence-associated β-galactosidase (SA-β-gal) activity by X-gal staining

3T3 cells transduced and G418-selected to express tet-inducible *Tcf7l1* were seeded in 100,000 cells/35 mm well and retroviral infected with pBabe-puro-*HRAS*$^{G12V}$ or pBabe-puro vector the next day. Puromycin selection and doxycycline treatment for transgene induction began 48 hr post infection. Cells were fixed in 4% formaldehyde for 3–5 min, washed, and stained in X-gal staining buffer (5 mM $K_3Fe(CN)_6$, 5 mM $K_4Fe(CN)_6$•3$H_2O$, 150 mM NaCl, 2 mM $MgCl_2$, 1 mg/mL X gal in citric acid/sodium phosphate buffer, pH 6.0) overnight at 37°C in dark. SA-$\beta$-gal activity was determined by quantification of blue cells after X-gal staining at pH 6.0.

## Flow cytometric analysis of tumor-infiltrating immune cells

Tumors derived from xenografted SCC were minced and enzyme digested in 0.25% Collagenase type 1A (Sigma-Aldrich) in HBSS at 37°C for 2–3 hr with gentle shaking (60 r.p.m.) and pipet mixing

every 15–20 min. After serial filtration through 70 nm and 40 nm mesh and neutralization with 10% FBS-containing media, cells were washed once with PBS-F (PBS with 2% FBS) and subjected to antibody staining against cell surface markers. Antibodies were used at the following concentration: FITC-conjugated anti-CD11b (1:200, BioLegend clone M1/70 #101206 RRID:AB_312789), PE-conjugated anti-Gr1 (Ly6GLy6C) (1:200; BD Biosciences clone RB6-8C5 #561084 RRID:AB_2034017), eFluor 780-conjugated anti-F4/80 (1:100, Thermo Fisher Scientific clone BM8 #47-4801-80 RRID:AB_2637188). After staining, cell were fixed with Cytofix/Cytosperm buffer (BD) and analyzed with a flow cytometer (LSRFortessa, BD Biosciences). FCS files were exported to be analyzed and graphed with FCS Express 6 Flow Cytometry (De Novo Software).

## FACS of murine adult skin and chemical-induced skin tumors

To enrich for epithelial cells from normal adult skin, dermal fat was scraped off from the backskin and the dermis was removed by digestion in 0.25% Collagenase type 1A (Sigma-Aldrich) in HBSS at 37°C for 30 min to 2 hr with gentle shaking (60 r.p.m.). The epidermal sheet was then rinsed with PBS, minced, and incubated with 0.25% EDTA-free Trypsin at 37°C for 30 min with gentle shaking (60 r.p.m.) to release basal keratinocytes and hair follicles. Cells were filtered through 70 nm and 40 nm mesh into 10% chelated FBS-containing media for neutralization. Filtered cell suspensions were pelleted at 400 xg for 10 min and live cells were FACS sorted based on DAPI staining.

To enrich for epithelial cells from DMBA/TPA-induced papilloma and SCC samples, tumors were minced and digested in 0.25% Collagenase type 1A (Sigma-Aldrich) in HBSS at 37°C for 2–3 hr with gentle shaking (60 r.p.m.) and pipet mixing every 15–20 min. Cells were filtered through 70 nm and 40 nm mesh into 10% chelated FBS-containing media. Residues were further trypsinized at 37°C for 30 min, filtered into neutralization media, and combined with the first batch. After centrifuge and resuspension, live cells were FACS sorted based on DAPI staining.

## Statistical methods

All animal experiments were done with the number of mice indicated in each experiment. The sample size in the DMBA/TPA tumorigenesis experiment was determined using Fisher's Exact Test to achieve power of 85% and an alpha of 0.05. Data were analyzed for statistical significance using either two-tailed Student's t-test or one-way ANOVA as indicated. Statistical significance in tumor incidence was determined by log-rank test using Prism software. Error bars denote standard deviation (s.d.).

## Acknowledgements

This work was supported by a research grant from the Texas Cancer Prevention and Research Institute RP110153 (to HN), and NCI 7R01CA194617, 7 R01 CA194062 and T Boone Pickens Endowment (to KYT). AK was supported by NIH T32GM088129-03 (to Dr. M K Estes), NIH T32HL92332-11, T32HL92332-12 (to Dr. H Heslop), and the Howard Hughes Medical Institute Med into Grad Initiative. JH was supported by NIH T32HL092332-07 (to Dr. H Heslop) and CPRIT RP101499 (to Dr. J Rosen). This project was also supported by the Cytometry and Cell Sorting Core at BCM with funding from the National Institutes of Health (P30AI036211, P30CA125123 and S10RR024574). STR DNA fingerprinting was done by the CCSG-funded Characterized Cell Line Core at MD Anderson Cancer Center, NCI # CA016672.

## Additional information

### Funding

| Funder | Grant reference number | Author |
| --- | --- | --- |
| Cancer Prevention and Research Institute of Texas | RP110153 | Hoang Nguyen |
| Cancer Prevention and Research Institute of Texas | RP101499 | Jeffrey M Howard |
| National Institutes of Health | T32-HL092332-07 | Jeffrey M Howard |

| National Institutes of Health | T32HL92332 | Amy T Ku |
|---|---|---|
| National Institutes of Health | T32GM088129 | Amy T Ku |
| National Institutes of Health | 7R01CA194617 | Kenneth Y Tsai |
| National Institutes of Health | R01 CA194062 | Kenneth Y Tsai |
| T. Boone Pickens Endowment | | Kenneth Y Tsai |

The funders had no role in study design, data collection and interpretation, or the decision to submit the work for publication.

### Author contributions

ATK, Conceptualization, Data curation, Formal analysis, Validation, Investigation, Visualization, Methodology, Writing—original draft, Writing—review and editing; TMS, Conceptualization, Investigation, Visualization, Methodology, Writing—review and editing; ASR, Formal analysis, Investigation, Visualization; JMH, DNC, Resources; CNR, QM, GG, DL, DY, VC, Investigation; MB, Supervision; AHD, Resources, Visualization; KYT, Resources, Supervision; HN, Conceptualization, Formal analysis, Supervision, Funding acquisition, Investigation, Methodology, Writing—original draft, Writing—review and editing

### Author ORCIDs

Hoang Nguyen, http://orcid.org/0000-0002-1091-7483

### Ethics

Animal experimentation: All mice were maintained in the AALAC-accredited animal facilities at Baylor College of Medicine and MD Anderson and all mouse experiments were conducted according to protocols approved by committees at Baylor College of Medicine (AN-4907) and MD Anderson (ACUF00001396-RN00).

# Additional files

### Supplementary files

• Supplementary file 1. Differentially-expressed gene lists and associated pathway analyses of human SCC (SCC13) cells overexpressing TCF7L1 or TCF7L1 deletion mutants. Tab one shows the list of genes that are differentially expressed between SCC13 cells overexpressing TCF7L1 or TCF7L1 mutants and cells expressing control vector. For genes in the datasets, HUGO symbol, Cufflinks XLOC identifier and locus are indicated. In the differentially expressed gene list, fold changes were computed after anti-log2-transformation, and t statistics (p values and FDR-adjusted q values) were computed for the corresponding coefficient and the significance was determined by q < 0.05. Tab two shows the genes represented in the heat map. The heat map with unsupervised clustering was constructed with 500 genes that have the largest coefficient of variation based on batch-corrected FPKM counts (fragments per kilobase of transcript per million mapped reads), and scaled over geometric mean. The red and green indicate expression values above or below the mean expression, respectively. Tab three and four respectively show a list of GO term biological process and Kyoto Encyclopedia of Genes and Genomes (KEGG) pathways that were derived from significantly induced or repressed gene lists with a twofold difference by the overexpression of TCF7L1 and TCF7L1ΔN but not TCF7L1*. Top 100 altered pathways were listed by p-value ranking.

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
