## [Decision Letter]

Thank you for submitting your article "TCF3 promotes skin tumorigenesis independently of β-catenin through induction of LCN2" for consideration by *eLife*. Your article has been reviewed by two peer reviewers, and the evaluation has been overseen by a Reviewing Editor and Fiona Watt as the Senior Editor. The reviewers have opted to remain anonymous.

The reviewers have discussed the reviews with one another and the Reviewing Editor has drafted this decision to help you prepare a revised submission.

Summary:

This manuscript examines the role of Tcf3 in cancer by focusing on effects of Tcf3 expression during the development and progression of squamous cell carcinoma using mouse and human systems. The authors strongly and clearly show that the β-catenin interaction domain of Tcf3 is not necessary for its effects on skin tumor promotion. Through RNA-seq analysis from xenografts overexpressing different mutant forms of Tcf3, the authors attribute the effects observed to induction of Lipocalin-2. This manuscript has important findings relating to the role of Tcf3 and its interaction with the Wnt signaling pathway, and upon the necessary revisions, can prove interesting for the readership of *eLife*.

Essential revisions:

1) The manuscript does not sufficiently address or discuss how Tcf3 could cause the effects observed in skin but appear to be different in breast cancer? Is the repressor-only effect of Tcf3 something specific to SCC, or is it indicative of a mechanism broadly relevant to cancer in other organs? Can the authors experimentally test this or add a discussion of these two organ systems?

2) The manuscript starts off showing that Tcf3 overexpression in a GEMM using the K14 promoter. However, the authors do not make use of the advantages of such a GEMM in the rest of the manuscript. All the follow up experiments are based on xenograft models in immune-deficient mice. The cell lines are extremely useful for determining the function of their different Tcf3 mutants, but it would be important to analyze the data in the original mouse model. For example, is Lcn-2 upregulated in papillomas and SCCs in the Tcf-3 induced mice? Would the tumor incidence be reduced, if Lcn-2 was deleted in this GEMM? Would turning off Tcf3 by removing Dox lead to decrease of tumorigenesis?

3) The authors show that Tcf3 induction can lead to tumor development with DMBA treatment only. This would suggest that the main role of Tcf3 is in tumor promotion, likely by replacing the inflammatory action of TPA. The authors should analyze the immune compartment of these tumors, and determine whether the Tcf3-induced tumors have increased inflammatory infiltrates and mediators to account for the increased tumor burden. IL-22 has been shown to link inflammation to keratinocyte proliferation, thus, it should be measured in the Tcf-3 induced tumors.

4) Lcn-2 deletion has been shown to decrease neutrophil infiltrate in a chemically-induced psoriasis-like disease model (Aldara treatment) (Shao S et al., 2016, JID). Aldara is used as a treatment for SCCs in patients, thus it would be interesting to determine whether Lcn-2 deletion in the setting of skin tumors leads to depletion of neutrophils.

5) Can authors determine whether the staining of LCN-2 and TCF3 correlate in human SCCs? In other words, do the cells expressing TCF3 also express LCN-2?

---

## [Author Response]

*Essential revisions:*

*1) The manuscript does not sufficiently address or discuss how Tcf3 could cause the effects observed in skin but appear to be different in breast cancer? Is the repressor-only effect of Tcf3 something specific to SCC, or is it indicative of a mechanism broadly relevant to cancer in other organs? Can the authors experimentally test this or add a discussion of these two organ systems?*

We agree that on the surface it seems that TCF7L1 may function differently in breast cancer and skin SCC based on the observation that TCF7L1 has different activity on WNT reporter in vitro. However, there are no data demonstrating that TCF7L1 acts differently in breast and skin in vivo. Our work is in fact the first to use deletion mutations of TCF7L1 to show that TCF7L1 does not require binding to β-catenin to promote tumor growth. It is possible that this repressor function of TCF7L1 may be similarly required in other types of cancer besides skin SCC, but that remains to be determined experimentally in vivoin those other cancer models.

Although TCF7L1 plays a tumorigenic role in xenograft models of both breast and colorectal cancer, no experiments were conducted to identify which function of TCF7L1 is required for its tumorigenic role in these two models. In colorectal cancer cell line HCT116, downregulation of TCF7L1 increases WNT reporter TOP- FLASH activity in vitro, suggesting that TCF7L1 functions as a repressor. However, downregulating TCF7L1 in breast cancer cells decreases expression of some WNT-responsive genes and increases expression of others, implying that TCF7L1 functions as an activator or repressor depending on its targets.

Although TCF7L1 may function as an activator or repressor on different sets of target genes in breast cancer cells, its tumorigenic role may depend only on its function as a repressor. Whether the repressor function of TCF7L1 is broadly required in its tumorigenic role will need to be demonstrated in different cancer models using the similar deletion mutations of TCF7L1 that we have used.

We have revised the Discussion section to include the above points.

*2) The manuscript starts off showing that Tcf3 overexpression in a GEMM using the K14 promoter. However, the authors do not make use of the advantages of such a GEMM in the rest of the manuscript. All the follow up experiments are based on xenograft models in immune-deficient mice. The cell lines are extremely useful for determining the function of their different Tcf3 mutants, but it would be important to analyze the data in the original mouse model. For example, is Lcn-2 upregulated in papillomas and SCCs in the Tcf-3 induced mice?*

Thank you for these comments, we now include data showing that expression of LCN2 is undetectable in normal skin but induced in both mouse and human skin SCC (new Figure 10—figure supplement 2). Moreover, overexpression of TCF7L1 in skin induces expression of LCN2 (new Figure 10—figure supplement 1). However, once tumors are formed, overexpression of TCF7L1 does not further increase the already upregulated level of LCN2 in the tumors.

We have included the above findings in the new supplementary figures and revised the Discussion section as follows:

“Consistent with its proposed role in skin SCC, expression of LCN2 is undetectable in normal epidermis and is robustly upregulated in both murine and human skin SCC (Figure 10, Figure 10—figure supplement 2). […] Since LCN2 is upregulated in TCF7L1-induced skins but not upregulated any higher in TCF7L1-induced tumors, it suggests that factors regulating the expression of LCN2 including TCF7L1 may already be in excess in the tumors; hence further increases of TCF7L1 do not affect expression of LCN2.”

*Would the tumor incidence be reduced, if Lcn-2 was deleted in this GEMM?*

We are aware that the xenograft model can be used only to dissect whether TCF7L1 requires LCN2 to promote growth of xenografted SCC cells and cannot address whether TCF7L1 requires LCN2 to increase tumor incidence, which could only be addressed with the DMBA/TPA model using GEMM. However, the GEMM experiment requires the intercrossing the tet-inducible *Tcf7l1* line, which is in FVB/N background, and *Lcn2* null mice, which are in C57BL/6 background. Since mice in C57BL/6 background have a different propensity for tumorigenesis, we would need to backcross the line with the FVB/N line before using the mice in the DMBA/TPA induced skin SCC model. Given that multiple crosses and generations are required to obtain the desired genotype, it would take more than 2 years to receive results from experiments conducted in tet-inducible *Tcf7l1* mice in the background of *Lcn2* null line. For this reason, we used the xenograft model to dissect TCF7L1’s requirement for LCN2 in its tumor-promoting role, while understanding its limitation.

In order to identify which stage of tumorigenesis affected by TCF7L1 requires induction of LCN2, we would need to inducibly ablate *Lcn2* in the tet-inducible *Tcf7l1* mouse model at different stages of tumor development (pre- and post-papilloma formation, pre- and post-SCC). Although a new *Lcn2*^fl/fl^ line has recently been generated (Mosialou et al., Nature 2017) which we hope to use in the future, accomplishing this is beyond the time frame for the revision of this manuscript.

*Would turning off Tcf3 by removing Dox lead to decrease of tumorigenesis?*

We agree that results from this experiment would reveal whether sustained overexpression of TCF7L1 is required to sustain tumor growth, progression into SCC or that it contributes to only papilloma development. Results from this experiment will provide additional insights into the role of TCF7L1 in tumorigenesis process; however, including the breeding time this experiment requires at a minimum 36 weeks, which is outside the time frame (2-3 months) allowed for resubmission. The proposed experiment requires putting 8 week old mice on dox-containing or normal diet and then subjecting them to DMBA/TPA treatment. And then after tumors are formed at different time points post DMBA/TPA treatment, half of the dox-diet group will be put back on a normal diet with continued TPA treatment for additional weeks until the end point of the experiment.

*3) The authors show that Tcf3 induction can lead to tumor development with DMBA treatment only. This would suggest that the main role of Tcf3 is in tumor promotion, likely by replacing the inflammatory action of TPA. The authors should analyze the immune compartment of these tumors, and determine whether the Tcf3-induced tumors have increased inflammatory infiltrates and mediators to account for the increased tumor burden. IL-22 has been shown to link inflammation to keratinocyte proliferation, thus, it should be measured in the Tcf-3 induced tumors.*

As advised, we immunostained uninduced and TCF7L1-induced skins that were treated with vehicle control, DMBA, or DMBA/TPA. We did not observe any obvious changes in the amount of neutrophils and macrophages between the control and TCF7L1-induced skins under all 3 treatments. However, using flow cytometry analysis of xenografted dissociated tumors, we did observe increased neutrophil infiltration in TCF7L1-overexpressing tumors (Figure 10—figure supplement 2). We have included these findings in the revised result and Discussion sections.

As suggested by the reviewers, we also examined expression of IL-22 in the TCF7L1-induced skins and did not observe any difference between the control and induced group. Indeed IL-22 is secreted by immune cells and has been reported to affect keratinocyte proliferation and migration; however, IL-22 itself is not expressed in epidermal cells. Our RNA-seq confirms that the expression of IL-22 is undetectable in SCC cells and unaffected by overexpression of TCF7L1.

*4) Lcn-2 deletion has been shown to decrease neutrophil infiltrate in a chemically-induced psoriasis-like disease model (Aldara treatment) (Shao S et al., 2016, JID). Aldara is used as a treatment for SCCs in patients, thus it would be interesting to determine whether Lcn-2 deletion in the setting of skin tumors leads to depletion of neutrophils.*

We grafted SCC cells that were engineered to express vector control or *Tcf7l1* with nonspecific shRNAs or shRNAs against *LNC2*. As expected, TCF7L1-overexpressing tumors were larger than control tumors, but the TCF7L1-overexpressing tumors were smaller when co-expressing shRNAs against LCN2. Through flow cytometry analysis of dissociated tumors, we found that overexpression of TCF7L1 resulted in increased neutrophil infiltration but downregulation of LCN2 in the TCF7L1-overexpressing tumors did not impinge on TCF7L1’s ability to increase neutrophil infiltration. This result leads us to conclude that while TCF7L1 requires LCN2 to promote tumor growth, it works independently of LCN2 to stimulate neutrophil infiltration. These results also show the complexity of the role of TCF7L1 in the tumorigenic process and make our profiling experiments valuable for further future mechanistic dissections.

We have included the new finding in the new Figure 10—figure supplement 2 and revised the Results section as follows:

“Since LCN2 has been shown to stimulate neutrophil chemotaxis in vitro(Shao et al. 2016), and inflammation facilitates tumor growth, we evaluated whether overexpressing TCF7L1 increases neutrophil infiltration in tumors and whether downregulating LCN2 alters their infiltration. […] Together, our data therefore imply that TCF7L1 accelerates human SCC cell migration and tumor growth through induction of LCN2 but does not require LCN2 to stimulate neutrophil infiltration”.

*5) Can authors determine whether the staining of LCN-2 and TCF3 correlate in human SCCs? In other words, do the cells expressing TCF3 also express* LCN-2?

As recommended, we immunostained human skin SCC samples for LCN2 and TCF7L1 expression and found that while only 10 out of 16 samples expressed TCF7L1, 15 samples expressed LCN2. Although all of the TCF7L1-positive samples expressed LCN2, 83.3% of TCF7L1-negative samples also expressed LCN2. These results imply that induction of LCN2 in human skin SCC is not dependent solely on TCF7L1.

We have added these data in the new Figure 10—figure supplement 3 and included our interpretation in the revised Discussion section as follows:

“Given that LCN2 is induced in the majority of human skin SCC, and is expressed in both TCF7L1-positive and -negative samples (Figure 10—figure supplement 2), factors other than TCF7L1 must also upregulate LCN2 in tumors. Indeed, NF-κB (Shen et al. 2006; Li et al. 2009; Karlsen et al. 2010; Mahadevan et al. 2011), C/EBP (Shen et al. 2006; Glaros et al. 2012), and NFAT1 (Gaudineau et al. 2012) all have been shown to activate expression of LCN2 in response to various stress and injury stimuli.”